# A deep learning pipeline for mapping in situ network-level neurovascular coupling in multi-photon fluorescence microscopy

Matthew W Rozak[1,2], James R Mester[1,2], Ahmadreza Attarpour[1,2], Adrienne Dorr[2], Shruti Patel[2], Margaret Koletar[2], Mary E Hill[3], Joanne McLaurin[3,4,5], Maged Goubran[1,2,5]*†, Bojana Stefanovic[1,2,5]*†

[1]Department of Medical Biophysics, University of Toronto, Toronto, Canada; [2]Physical Sciences, Sunnybrook Research Institute, Toronto, Canada; [3]Biological Sciences, Sunnybrook Research Institute, Toronto, Canada; [4]Department of Laboratory Medicine and Pathobiology, University of Toronto, Toronto, Canada; [5]Hurvitz Brain Sciences, Sunnybrook Research Institute, Toronto, Canada

*For correspondence:
maged.goubran@utoronto.ca
(MG);
bojana.stefanovic@utoronto.
ca (BS)

†These authors contributed
equally to this work

Competing interest: The authors
declare that no competing
interests exist.

Reviewing Editor: Jason P Lerch,
University of Oxford, United
Kingdom

## eLife Assessment

This work describes a highly complex automated algorithm for analyzing vascular imaging data from two-photon microscopy. This tool has the potential to be extremely **valuable** to the field and to fill gaps in knowledge of hemodynamic activity across a regional network. The **solid** biological application provides a demonstration of their pipeline's capabilities and suggests intriguing hypotheses around prolonged vascular tone changes, but will need to be followed up by further experiments to be conclusively demonstrated.

**Abstract** Functional hyperemia is a well-established hallmark of healthy brain function, whereby local brain blood flow adjusts in response to a change in the activity of the surrounding neurons. Although functional hyperemia has been extensively studied at the level of both tissue and individual vessels, vascular network-level coordination remains largely unknown. To bridge this gap, we developed a deep learning-based pipeline that uses two-photon fluorescence microscopy images of cerebral microcirculation to enable automated reconstruction and quantification of the geometric changes across the microvascular network, comprising hundreds of interconnected blood vessels, pre and post-activation of the neighboring neurons. The pipeline's utility was demonstrated in the Thy1-ChR2 optogenetic mouse model, where we observed network-wide vessel radius changes to depend on the photostimulation intensity, with both dilations and constrictions occurring across the cortical depth, at an average of 16.1±14.3 µm (mean ± SD) away from the most proximal neuron for dilations; and at 21.9±14.6 µm away for constrictions. We observed a significant heterogeneity of the vascular radius changes within vessels, with radius adjustment varying by an average of 24 ± 28% of the resting diameter, likely reflecting the heterogeneity of the distribution of contractile cells on the vessel walls. A graph theory-based network analysis revealed that the assortativity of adjacent blood vessel responses rose by 152 ± 65% at 4.3 mW/mm$^2$ of blue photostimulation *vs.* the control, with a 4% median increase in the efficiency of the capillary networks during this level of blue photostimulation in relation to the baseline. Interrogating individual vessels is thus not sufficient to predict how the blood flow is modulated in the network. Our pipeline, enables tracking of the microvascular network geometry over time, relating caliber adjustments to vessel wall-associated cells' state, and mapping network-level flow distribution impairments in experimental models of disease.

## Introduction

To support healthy brain functioning, the cerebrovascular network undergoes continual adjustments in vessel calibers (*Iadecola, 2017*; *Kisler et al., 2017*; *Kenney et al., 2016*). Neurovascular coupling refers to the change in blood flow following changes in the level of neuronal activity: under physiological conditions, a generous buffer of nutrients is granted to activated parenchyma via the capillary network (*Iadecola, 2017*; *Phillips et al., 2016*). This buffer is maintained through finely tuned regulation of flow through changes in the vessel caliber, mediated via contractile cells in the vessel walls. In the absence of such tuning, pockets of tissue could experience inadequate access to metabolites (*Secomb et al., 2000*). Alterations in smooth muscle cells, pericytes, and astrocytes may lead to compromises in vessels' dilatory capacity and thus deficits in neurovascular coupling (*Hartmann et al., 2021*; *Hall et al., 2014*; *Mester et al., 2021*; *Adams et al., 2018*). In various brain pathologies, including Alzheimer's disease, stroke, and trauma, regional blood flow regulation gets impaired through vessel loss and/or dysfunction of the vessels' dilatory capacity, resulting in regions of ischemia/hypoxia (*Carroll et al., 2020*; *Yang et al., 2022*). Previous studies have examined either individual vessels or the tissue level responses, with little attention having been paid to the vascular network, though network dysfunction frequently is associated with accelerated disease progression and long-term symptomatology (*Kenney et al., 2016*; *Mayer et al., 2011*; *Park et al., 2009*; *Petkus et al., 2016*; *Ramos-Cejudo et al., 2018*; *Franzmeier et al., 2019*; *Rabi, 2019*; *Boehme et al., 2021*; *Koliatsos and Rao, 2020*; *Ware et al., 2020*).

While there is copious data on the functioning of individual vessels, interrogation of the microvascular network remains a challenge, in terms of both data acquisition and analysis (*Hartmann et al., 2021*; *Hall et al., 2014*; *Hill et al., 2015*; *Lindvere et al., 2010*; *O'Herron et al., 2022*). To date, studies on the brain vasculature have been done by sparsely imaging individual blood vessels at the cellular scale (*Hartmann et al., 2021*; *Hall et al., 2014*; *Hill et al., 2015*; *Alarcon-Martinez et al., 2020*; *Sakadžić et al., 2011*; *Guo et al., 2023*; *Giblin et al., 2023*; *McDowell et al., 2021*; *Kim et al., 2023*; *Mester et al., 2019*; *Kim et al., 2012*; *Kleinfeld et al., 1998*), thereby severely undersampling the microvascular network; or by evaluating the averaged flow over many vessels at the mesoscopic scale, thus failing to discern the flow through individual vessels. A critical gap in the field is the characterization of flow across hundreds of individual vessels, while imaging the network structure that links them together to determine how the vascular response is coordinated across the network. This gap is particularly significant as studies investigating blood flow across several vessels at a time (imaged by varying the line acquisition pattern *Hartmann et al., 2021*; *Alarcon-Martinez et al., 2020*) have shown highly heterogeneous responses among capillaries. Neuronal function impairments arise wherever local metabolite supply becomes inadequate, notwithstanding the physiological level of flow across the network as a whole, making mapping of vessel changes across the network of particular importance.

To address the limitations of previous work, we developed a novel deep learning (DL)-based pipeline for mapping changes to the geometry of the brain vascular network following neuronal activation, from a time series of volumetric two-photon fluorescence microscopy (2PFM) data. Neuronal activation was elicited by photostimulation of pyramidal neurons expressing Channelrhodopsin-2 (ChR2; *Boyden et al., 2005*) in the Thy1-ChR2-YFP mouse model (*Arenkiel et al., 2007*). Our DL pipeline enabled automatic and accurate segmentation, registration, and network analysis of large 2PFM datasets across time. We applied our pipeline in a dataset of 17 Thy1-ChR2-YFP mice to map photostimulation-induced changes across the microvascular network - at the level of individual vessels and at the level of vertices spaced every micrometer along the vessels - in relation to the distance to the closest pyramidal neurons expressing the optogenetic actuator and across the cortical depth. Our findings demonstrate the utility of our pipeline for studying in situ microvascular morphology and function to address various neuroscientific hypotheses.

## Methods

### Cohort

#### Animals

All experimental procedures in this study followed the ARRIVE 2.0 guidelines (*Percie du Sert et al., 2020*). They were approved by the Animal Care Committee of the Sunnybrook Research Institute

(Animal use protocols 20169, 21619, 22619, 23619), which adheres to the policies and guidelines of the Canadian Council on Animal Care and meets all the requirements of the Provincial Statute of Ontario, Animals for Research Act, and the Canadian Federal Health of Animals Act. We used 15 male and 13 female Thy1-ChR2-YFP mice (#007612, line 18, Jackson Laboratory) at 6–12 months of age (283.9±67.0 days), weighing 28.8±7.1 g on the day of imaging. The mice were bred in-house and housed under a 12 hr light/dark cycle (*Arenkiel et al., 2007*). In this mouse line, Channelrhodopsin-2 is expressed preponderantly by pyramidal neurons with soma in cortical layer 5 and dendrites projecting to the cortical surface, enabling depolarization of their cellular membranes and action potential generation upon blue light illumination (*Mester et al., 2019*; *Arenkiel et al., 2007*). An attrition schematic is provided in *Appendix 1—figure 1*.

## Surgical preparation

On the day of imaging, mice were induced with 5% isoflurane, transferred to a rectal probe feedback-regulated heating pad (CWE Inc, Ardmore PA) to maintain a temperature of 37 °C, and maintained under 2% isoflurane. A subcutaneous injection of 1 mL of lactated Ringer's solution was administered at the start of surgery. Throughout the surgical preparation and imaging, we monitored breath rate, heart rate, arterial blood oxygen saturation, pulse distention, and breath distention via a pulse oximeter, with a probe mounted on the thigh (MouseOx, STARR Life Sciences; *Supplementary file 1, table 1*). For fine control over respiration, the mice were tracheostomized with an endotracheal tube (20 Ga catheter) and ventilated with a gas mixture of 20–30% $O_2$ and medical air using a small animal ventilator (SAR 830 /P, CWE Inc, Ardmore PA) set to 115–130 breaths per minute at an inspiratory/expiratory ratio of 1:3–4. Their heads were immobilized via ear bars during the placement of the cranial window and imaging. The tail vein was cannulated with a 26 Ga catheter to enable fluorophore and anesthetic delivery. A cranial window was implanted over the forelimb region of the primary somatosensory cortex (AP 0.25 mm, ML 2.0 mm); the dura was excised, and 1% agarose was applied between the brain and the glass coverslip. A well was built using dental cement to allow water immersion of the objective. Texas Red dextran (70 kDa MW, Thermo Fisher Scientific Inc, Waltham MA) was diluted in PBS and injected through the tail vein catheter at a concentration of 33 mg/kg for a total injection volume between 0.1 and 0.2 mL. A line containing 0.01 g/mL alpha chloralose was connected to the tail vein catheter. The animal was then positioned underneath the microscope objective, and the structural 2PFM scan acquired, as detailed below. Seventeen mice (nine male and eight female) made it through the surgical preparation (*Appendix 1—figure 1*). Following the structural acquisition, the isoflurane was discontinued, and the continuous infusion of alpha chloralose commenced at 40 mg/kg/hr.

## Imaging

### Two-photon fluorescence imaging and optogenetic stimulation

Mice were imaged on an FVMPE-RS microscope (Olympus, Japan) using a 25 x/1.05NA objective. An Insight tunable Ti:Sapphire near-infrared laser (SpectraPhysics, USA) was used for 900 nm excitation of Texas Red-labeled vasculature and YFP-labeled ChR2-expressing pyramidal neurons. Two visible light stimulation lasers were used to excite ChR2 at either 458 nm or 552 nm (control). The optical setup is shown in *Figure 1*. The imaging field of view was selected to include at least one penetrating artery and vein yet avoid major pial vessels and thus signal loss in the underlying cortex. Structural scans were acquired under 2% isoflurane using a Galvano scanner, with 2 x averaging, a z-step of 0.99 µm, and a nominal lateral resolution of 0.99 µm. Following structural scanning, mice were allowed to rest for 5 min to let blood flow equilibrate from brain exposure to room light. The five-minute resting period was observed due to prior reports of red blood cell velocities following optogenetic ChR2 stimulation in the same mouse strain remaining altered for a minute following photostimulation and due to potential pericyte-mediated responses via cytoskeleton reorganization occurring over a minute (*Hartmann et al., 2021*; *Mester et al., 2019*).

Functional scans were acquired over baseline and post-stimulus periods, flanking a period of blue light illumination. The hemodynamic response to optogenetic stimulation in the Thy1-ChR2-EYFP mouse model was previously observed to last about 45 s and observed to last for minutes in our experimental model (*Appendix 1—figure 2*; *Mester et al., 2019*). These vascular radius time courses were similar to data collected in other optogenetic models of mural cell stimulation where arteries were

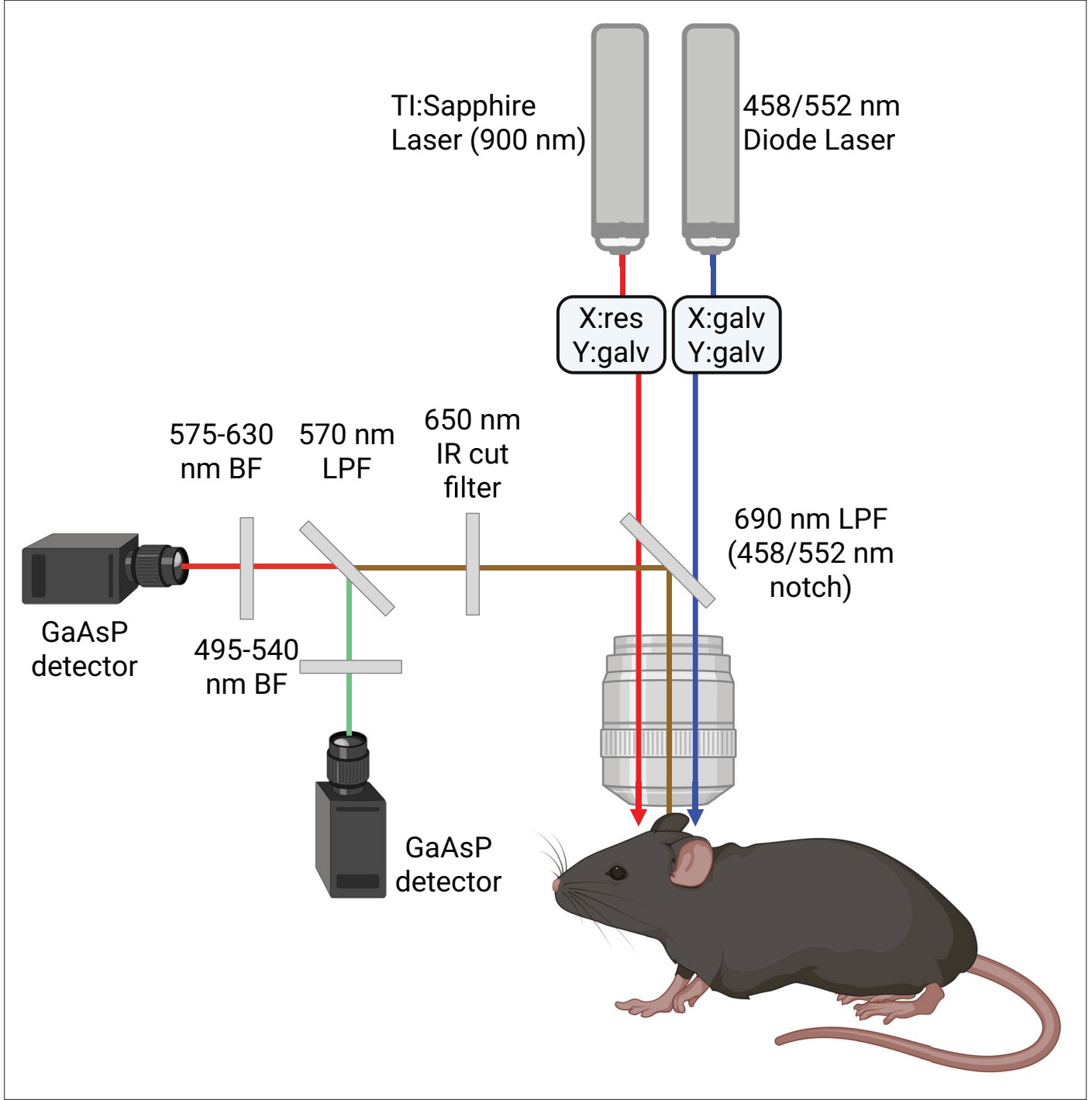

**Figure 1.** Photostimulation setup. The excitation and stimulation light pass through a FV30-NDM690 dichroic mirror with two notch filters, at 458 nm and 552 nm, to excite TexasRed, EYFP, and ChR2 within the mouse. The emitted light passes through the objective, is reflected off the FV30-NDM690 dichroic mirror, and passes through a 650 nm barrier filter before reaching a 570 nm long pass filter (LPF) separating emitted light from EYFP and TexasRed, which respectively pass through 495–540 nm and 575–630 nm barrier filters to be collected via GaAsP detectors.

observed to dilate for approximately 60 and 20 s respectively, and capillaries were observed to dilate on the scale of minutes (*Hartmann et al., 2021*; *O'Herron et al., 2022*). We thus constrained our volumetric acquisitions to 45 s. Each volumetric scan lasted for 42.98 s and used the resonant scanner with 5 x frame averaging (resulting in a signal-to-noise ratio on the vasculature channel of 6.9±1.6), a nominal lateral resolution of 0.99 µm, a z-step of 2.64 µm, and traversed 250.8 µm of cortical depth.

This z-step was chosen because larger z-steps led to discontinuities in the capillary bed segmentation. We imaged the brain in sections from the cortical surface to a depth of 250 μm and from 250 μm to 500 μm of cortical depth. The acquisitions were split into two slabs to maximize the volume coverage without creating discontinuities in the capillary segmentation masks while maintaining a scan duration of under 45 s. Stacks were acquired either from the cortical surface down or from the bottom of the stacks toward the cortical surface on different paired acquisitions, so that the post-stimulus delay of image acquisition at different cortical depths was not constant. A 239 μm diameter circular photostimulation region was positioned 250 μm beneath the cortical surface. The photostimulation light was raster scanned over this ROI for 5 s between imaging scans, with a pixel dwell time of 4 μs for each pixel of the stimulation ROI. Each pixel in the photostimulation ROI was thus excited every 501.55 ms, that is at about 2 Hz. We used two laser powers for 458 nm photostimulation: 1.1 mW/mm$^2$ and 4.3 mW/mm$^2$. The 1.1 mW/mm$^2$ is just above the threshold for ChR2 photoactivation and is expected to elicit a small degree of neuronal activity, whereas the 4.3 mW/mm$^2$ photoactivation elicits more neuronal activity while still being below ChR2 saturation level (*Boyden et al., 2005*; *Lin et al., 2009*). Low powers were used to activate ChR2-expressing neurons to minimize tissue heating (*Rungta et al., 2017*). We also performed control experiments via 552 nm photostimulation (where ChR2 excitation falls to 4% of its peak) at 4.3 mW/mm$^2$ (*Lambert, 2019*). Mice rested for approximately 5–7 min between scan pairs for the vascular tone to return to its resting state. Photostimulation parameters were presented in randomized order for each mouse.

## Segmentation and graph extraction

### Ground truth generation

To generate ground truths, we used ilastik's pixel classification workflow, which relies on random forests, to annotate blood vessels and pyramidal excitatory neurons somas in 42 volumes from 25 Thy1-ChR2-EYFP mice for training our DL model (24 images from 16 Thy1-ChR2-EYFP mice from other studies were included in the training, validation, and test cohorts to increase our dataset size). Images were semi-automatically segmented in groups of up to four each, with a size of 507 x 507 × 250 μm each, due to ilastik's inability to handle large amounts of annotations over more than a few volumes. The rater labeled targets (neurons vs. blood vessels), corrected mistakes, and classified the pixels where the output was uncertain (as shown using ilastik's uncertainty guidance feature). During the manual annotation, feature selection was repeatedly optimized using ilastik's suggest feature function (wrapper method), leading to different features being used for the random forest model for each set of images. Small (under 50 pixels) isolated vascular components were removed with connected component analysis in Python using scikit-image and connected-components-3d. Testing data was withheld until the final model was selected based on the models' performance evaluation on the validation data set.

### Data preprocessing and augmentation

We used the MONAI python package for data preprocessing, augmentation, model training, and prediction (*Consortium, 2022*). All images and ground truth segmentation masks were up-sampled to an isotropic voxel size using bilinear and nearest neighbor interpolation. Raw 10-bit image intensities were normalized to range from 0 to 1.0. Each volume had eight 128 x 128 × 128 pixel patches randomly cropped out for training. Data augmentation, transformations, and parameters are listed in *Supplementary file 2, table 2*. Spatial transformations were selected to expand data variety via cropping, rotations, and mirroring, hence exposing the network to images that would be acquired on different positioning of the animal under the microscope. Zooming and deformation transformations were included to expose the network to small changes in the size and morphology of the vasculature. Intensity and Gaussian transformations exposed the network to signal intensity and contrast variations. Dropping pixels was included as the resonant scanner acquisition occasionally yields images with some relatively low signal pixels.

### Model architecture

We trained a state-of-the-art 3D vision transformer (UNETR) model and the U-Net model for baseline comparison, as implemented in PyTorch using the MONAI library (*Consortium, 2022*). UNETR is a U-Net style architecture with a transformer-based multi-attention head encoder and a CNN decoder

(*Hatamizade, 2021*; *Kerfoot et al., 2019*). The encoder takes an input image (or patch from each batch) and breaks it down into a sequence of non-overlapping patches, each of size 16 x 16 × 16 pixels, which are weighted differently to account for variation in signal intensities and patterns within an image. The sequence of patches is then passed through a multi-head self-attention and multilayer perceptron encoder to capture self-attention between different pixels of the patch to encode long-range relationships between patches. The encoder is then connected to a CNN decoder with skipped connections to the encoder to map features back onto the original image at multiple spatial scales (*Hatamizade, 2021*). The model was trained on two-channel images (EYFP-expressing neurons and Texas Red-labeled vasculature) as inputs to segment neurons and vasculature. We chose the following parameters for the UNETR architecture: a 12 multi-attention head encoder, 16 convolutional features in the first layer of the encoder, a hidden layer size of 768, a multilayer perceptron size of 3072, and Monte Carlo dropout, where on each run of the model 10% of weights were zeroed (*Hatamizade, 2021*; *Kerfoot et al., 2019*; *Srivastava et al., 2014*).

The U-net model used is based on the original 2D U-net proposed by Ronneberger et al. in 2015 and extended into 3D via 3D convolutional operations and features residual blocks, parametric rectifying linear units, and instance normalization as described by *Kerfoot et al., 2019*; *Mojiri Forooshani et al., 2022*; *Ulyanov et al., 2017*; *He et al., 2015*; *He et al., 2016*; *Zhang et al., 2018b*. The residual blocks were implemented for better gradient flow during the network training. At the same time, the parametric ReLU activation functions were used to improve the ability of the model to adjust its weights during training. Instance normalization was implemented to reduce contrast differences between images fed into the network to improve model robustness. The model was implemented with five layers featuring 16, 32, 64, 128, and 256 channels and dropout. The U-net model was trained using the same data augmentation employed during the UNETR model training.

## Model hyperparameter optimization, training, and prediction

We used a grid search to determine optimal hyperparameters for the UNETR and UNet models within the following parameter space (*Mojiri Forooshani et al., 2022*; *Goubran et al., 2020*): loss functions (Dice Loss, Dice + Cross Entropy Loss, Dice +Focal Loss, or Tversky Loss), dropout rates (0.1, 0.2, or 0.3), learning rates (5e-3, 1e-3, 5e-4, 1e-4, or 1e-5), and the number of residual units for the Unet model (2 or 3; *Salehi et al., 2017*; *Lin et al., 2018*). We utilized the Adam optimizer (*Kingma and Ba, 2017*), and the best model during hyperparameter optimization was selected based on the validation Dice similarity coefficient (DSC) and trained for a maximum of 2400 epochs. The Dice score was the principal evaluation metric to maximize the overlap between ground truth and prediction masks. Precision and recall were used as secondary metrics to achieve balanced precision and recall where over-segmented results were produced. For early stopping, the epoch with the best performance of the DSC on the validation dataset was selected. The final model that had the best Dice score during hyper-parameter optimization was trained with a Dice + Cross Entropy Loss function, a dropout rate of 0.1, 2 residual units, a learning rate of 1e-5, and a batch size of 1 image with 8 crops per image. Training and optimization were performed on the Narval cluster of Calcul Quebec and the Digital Research Alliance of Canada, with each node using 498 GB of RAM and 4 Nvidia A100 GPUs, each with 40 GB HBM2 VRAM.

Ilastik utilized a random forest model from the Vigra library with the default parameters and 100 trees (*Berg et al., 2019*; *VIGRA Homepage, 2023*). We initially utilized all default 3D Color/Intensity/Texture/Edge features during ilastik feature selection and added 2D Color/Intensity/Texture/Edge features with a sigma of 20.0 pixels. These 2D features were added as a strong, largely planar artifact was observed towards the surface of the images in the neuron channel. We used the wrapper method for feature selection with a set size penalty of 0.10 to determine the optimal features from the starting set. The model was then trained using live updates on the Narval cluster of Calcul Quebec and the Digital Research Alliance of Canada, with each node using 249 GB of RAM and 2 x AMD Rome 7532 CPUs with 64 cores.

For the ilastik model, a subset of three training images from three mice was used as ilastik's random forest was unable to train a model with more data even on compute nodes with large memory and CPU resources. The model could never complete training within the time limits of the resource allocations. This inability to complete model optimization was expected as ilastik's documentation does not recommend training random forest models with full annotation datasets, and increasing the number

of annotations does not necessarily lead to better predictions for the pixel classification workflow (**Berg et al., 2019**). The ilastik random forest model was trained with whole images rather than with patches.

## Model comparison

To compare models, we employed several metrics evaluating the similarity between the ground truth and prediction: the Dice Similarity Coefficient (DSC), Precision, and Recall, as well as surface-based metrics: 95% Hausdorff distance (HD95) and mean surface distance. We specifically focused on mean surface distance and HD95 distance since the centerline extraction used in the downstream analysis was highly sensitive to surface irregularities. The model with the lowest standard deviation on the mean surface distance was to be selected absent statistically significant differences in the average mean surface distance between different models. Consistency in surface placement was key for extracting good centerlines and graphs from the segmentation masks. The metrics utilized are defined as follows:

$$DSC = \frac{2\,|X \cap Y|}{|X| + |Y|} \tag{1}$$

$$Precision = \frac{True\ Positives}{True\ Positives + False\ Positives} \tag{2}$$

$$Recall = \frac{True\ Positives}{True\ Positives + False\ Negatives} \tag{3}$$

$$HD95 = Max\left\{ Sup_{95\%,\,x \in X} d\,(x, Y)\,, sup_{95\%,\,y \in Y} d\,(X, y)\right\} \tag{4}$$

$$Mean\ Surface\ Distance = \frac{1}{N_X + N_Y}\left(\sum_{p=1}^{n_X} d\,(p, n_Y) + \sum_{q=1}^{n_Y} d\,(q, n_X)\right) \tag{5}$$

The DSC measures the overlap between the prediction and ground truth with a value of 1 corresponding to complete overlap, X represents the ground truth and Y represents the predicted value. Precision measures the rate of correctly returned predicted values, whereas recall assesses the rate of return of targeted values. For the Hausdorff 95% distance (HD95), the distance d is the infimum of the Euclidean distance between points $x$ and $y$ (from the surface of segmentation masks X and Y) to segmentation masks Y and X, respectively. HD95 computes the maximum of the 95th percentile of the supremums (*sup*) of these minimum surface distances (d(x,Y) and d(X,y)) between the boundaries of the ground truth and the predicted segmentation mask. The mean surface distance measures the average minimum distance between the boundary of the ground truth and the boundary of the predicted segmentation mask. In **Equation 5**, $n_x$ and $n_y$ are the sets of boundary points of X and Y, respectively, and $N_x$ and $N_y$ are total number of boundary points $n_x$ and $n_y$. The infimum of the minimum Euclidean distance, d, from boundary points p from $n_x$, and q from $n_y$ to sets $n_x$ and $n_y$ is then computed and averaged to compute the Mean Surface Distance. Together, the HD95 and Mean Surface Distance assess model performance at the boundaries of the segmentation masks.

## Graph extraction of cerebrovascular networks

Graph extraction was performed in Python except for the centerline calculation, which was performed in MatLab (R2021a). Upsampled image acquisitions (0.99 µm isotropic voxel size) were registered with ANTsPy using the rigid registration method with a total sigma of 2 pixels (for smoothing within the registration function) and mean squared error as the similarity metric as input parameters of the *registration* function (**Figure 2A and B**). We selected the first baseline scan from each region of interest that was scanned from the bottom to the top to serve as a reference to which all other images from the same region were aligned. The calculated transforms were used to transform images from the same ROI to the space of the reference image using linear interpolation (**Avants et al., 2014**). We then generated segmentation masks for each aligned image using our trained UNETR model with sliding window inference, and we retained Monte Carlo dropout during prediction to create an ensemble of 20 UNETR models, so as to assess the model uncertainty (**Figure 2D**; **Gal and Ghahramani, 2016**).

To extract the centerlines, we assumed that no background was present within the vessels, as regions of background within vessels disrupt the centerline extraction algorithm. Background-labeled

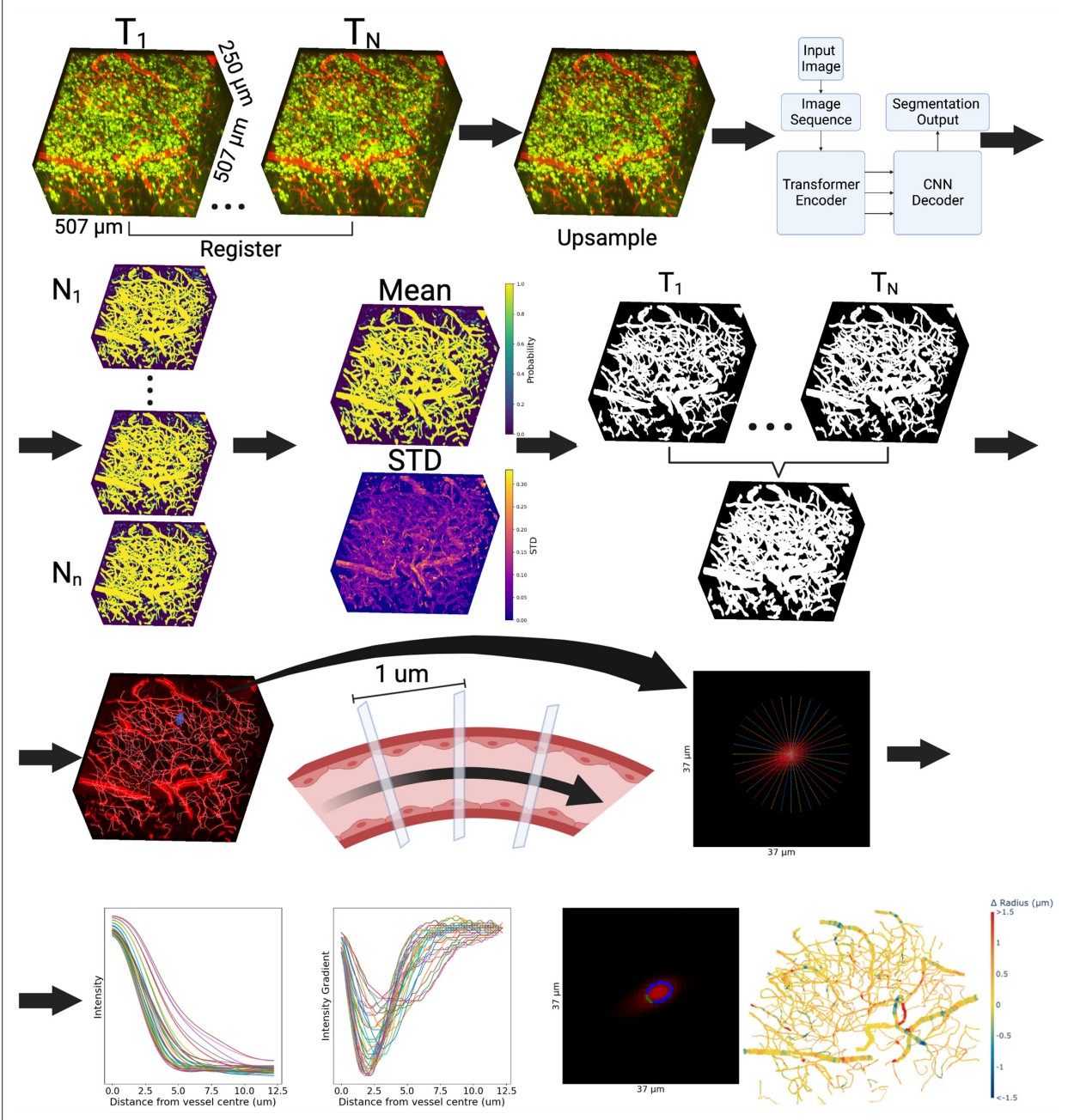

**Figure 2.** Computational analysis pipeline. (**A**) The stacks of 2PFM slices were registered using ANTS rigid registration and aligned to the reference time point. (**B**) Images were upsampled using bicubic interpolation to an isotropic resolution of 0.99 x 0.99 x 0.99 µm. (**C**) An ensemble of UNETR deep learning models with dropout generated segmentation masks at each time point, producing probability maps. (**D**) The mean and standard deviation of the probability of each pixel being vasculature were computed and used to create binary vascular segmentation masks. (**E**) The union over the vascular segmentation masks for all time points was computed, and background pixel clusters within vessel masks were removed. (**F**) The vascular segmentation mask was thinned down to centerlines and rendered as a graph, where edges were vessel segments connecting branch points (nodes). This skeleton was overlaid on the vasculature channel from which the neuron channel was subtracted. (**G**) The plane orthogonal to the tangent to the vessel's travel direction was computed every micrometer along the centerline. (**H, I**) 1D signal intensity profiles at each centerline vertex were computed in the orthogonal plane every 10°. (**J**) The boundary for each profile was placed at the minimum of the signal gradient for that signal intensity profile. (**K**) The raw intensity image with the detected boundary points, where outlier boundary points (in green) were defined as points over 2 standard deviations from the mean were excluded. (**L**) Visualization of the changes in vertex-wise radii on a sample vascular network.

pixels surrounded by blood vessel segmentations were thus filled in using connected component analysis. Disconnected vascular components under 50 pixels were assumed to be noise and removed. Each aligned binary segmentation mask was dilated three times with a footprint defined by scikit-image's disk function with a radius of 1 pixel. Next, we computed the union of all segmentation masks from the same ROI generated from registered images (*Figure 2E*). This common segmentation mask for all time points was eroded to a centerline via a medial axis transformation (*bwskel* function in Matlab [R2021a]). Critically, using the union of all time points minimized the sensitivity of centerline identification to red blood cell (RBC) stalls at individual time points. 'Hair' segments shorter than 20 µm and terminal on one end were iteratively removed, starting with the shortest hairs and merging the longest hairs at junctions with two terminal branches with the main vessel branch to reduce false positive vascular branches and minimize the amount of centerlines removed. This iterative hair removal functionality of the skeletonization algorithm is currently unavailable in Python, but is available in Matlab (*Lee et al., 1994*). The vascular centerlines were next used to construct vascular graphs (with *sknw*) for each ROI, rendering coordinates of vertices along the vessel's centerline as edges, and branch points as nodes (*Nunez-Iglesias et al., 2018*).

Based on the vascular graphs, we computed vascular radius estimates at each vertex at each time-point and then calculated the vertex's distance to the nearest YFP-expressing pyramidal neuron (as measured by the distance transformation at the vertex to the nearest labeled neuron, with the radius of the vessel subtracted off). We employed a two-stage approach to estimate the radius for each point on the centerline. First, using the binary segmentation mask, we calculated the distance from every vessel pixel in the mask to the nearest background pixel. These values were averaged for each vessel to estimate its radius. The radius estimate defined the size of the Gaussian kernel that was convolved with the 2d image slice to smooth the vessel: smaller vessels were thus convolved with narrower kernels.

In the second stage, the registered raw image was deconvolved with the point spread function of the microscope, as measured via FluoSpheres carboxylate-modified Microspheres (Cat# F8803, Thermo Fisher Scientific Inc, Waltham MA), using Richardson-Lucy deconvolution. To low-pass filter the centerline in advance of computing the tangent vector at each vertex, the coordinates of the vertices along the centerline were smoothed using a Gaussian with a sigma of 3. Next, the tangent vector to the centerline was specified by calculating the gradient of the centerline path. Given this local tangent to the vessel's direction of travel at a given vertex, we extracted image intensities in the orthogonal plane from the deconvolved raw registered image. A 2D Gaussian kernel with sigma equal to 80% of the estimated vessel-wise radius was used to low-pass filter the extracted orthogonal plane image and find the local signal intensity maximum searching, in 2D, from the center of the image to the radius of 10 pixels from the center. The orthogonal plane image was sampled every 10 degrees (as finer radial sampling did not improve the estimation *Appendix 1—figure 3*) along radial lines emanating from the local signal intensity maximum closest to the center of the image and 5 x bicubic upsampled to extract thirty-six 1D signal intensity profiles at that vertex, as shown in *Figure 2H*. These line profiles were then convolved with a 1D Gaussian kernel with a sigma of 80% of the estimated radius of that vessel, and the gradient of each profile was calculated as shown in *Figure 2I and J*. The vessel boundary was then placed at the local minimum of the gradient of each profile. The mean and standard deviation of the boundary distances for the 36 1D line profiles were calculated, and boundary points greater than 2 standard deviations away from the mean were excluded (*Figure 2K*). This radius estimation procedure was repeated for all vertices of all vessels.

In the last step, we computed the distance from each image voxel to the nearest YFP-expressing pyramidal neuron by computing the distance from every image pixel not belonging to the neuron mask to the closest YFP-expressing neuron's soma boundary (based on the neuron's binary segmentation mask). At the vessel vertices, these distances were adjusted by subtracting the vessel's radius. We thereby captured the distance from the vessel surface to the closest YFP-expressing neuron.

## Boundary detection validation

To assess the uncertainty in the radius estimates at each vertex, we simulated changes to the vascular diameter by 'resizing' the extracted orthogonal plane image and adding Gaussian noise to it. These steps were undertaken to validate the ability of the boundary detection method to estimate diameter changes and evaluate the robustness of the estimates to increased amounts of noise. For the diameter

change estimation, images were resampled from the orthogonal plane by a random factor, uniformly distributed in the range of 0.5x to 2x. Then, the aforementioned boundary detection methods were used to estimate vertex-wise caliber changes. The end goal of the assessment was to see how close the boundary detection method-based radius change matched the prescribed diameter change. In the second task, we added Gaussian noise with a sigma randomly chosen from a uniform distribution ranging from 0 to 500 SU to the orthogonal plane image. The noise was added to the image after deconvolution with the PSF and the extraction of the orthogonal plane but before Gaussian smoothing. We then reported on the percent change in the radius as a result of resizing or adding noise in relation to the baseline radius.

Second, our boundary detection algorithm was used to estimate the diameters of fluorescent beads of a known radius imaged under similar acquisition parameters. Polystyrene microspheres labeled with Flash Red (Bangs Laboratories, Inc, CAT# FSFR007) with a nominal diameter of 7.32 μm and a specified range of 7.32±0.27 μm as determined by the manufacturer using a Coulter counter were imaged on the same multiphoton fluorescence microscope set-up used in the experiment (identical light path, resonant scanner, objective, detector, excitation wavelength, and nominal lateral and axial resolutions, with 5 x averaging). The images of the beads had a higher SNR than our images of the vasculature, so Gaussian noise was added to the images to degrade the SNR to the same level of that of the blood vessels (SNR value of 5.05±0.15). The images of the beads were segmented with a threshold, centroids calculated for individual spheres, and planes with a random normal vector extracted from each bead and used to estimate the diameter of the beads. The same smoothing and PSF deconvolution steps were applied in this task. We then reported the mean and standard deviation of the distribution of the diameter estimates. A variety of planes was used to estimate the diameters.

## Vascular network analysis

Leveraging the graph representation of the vasculature in our pipeline, we next used graph theory to better understand the networks' behaviour upon neuronal activation. We looked at morphometrics including vascular segment count density, vascular length density, and mean vessel length for comparison with other work. To demonstrate the benefits of extracting a graphical representation of the vasculature in situ, we looked at graph theory metrics including the assortativity of vascular radius changes and changes to the global efficiency of the capillary (below 5 μm in radius) network following optogenetic stimulation. The assortativity measures the tendency for one vessel to dilate or constrict by a similar amount as its neighbors. The assortativity (q) of radius changes in response to stimulation is defined as the Pearson correlation coefficient of these changes on connected vessels:

Assortativity

$$q = \frac{\sum_{xy} xy \left( e_{xy} - a_x b_y \right)}{\sigma_a \sigma_b} \tag{6}$$

where $e_{xy}$ represents the fraction of vessels (edges) in the network that join together nodes with values x and y (i.e. radius changes with values x and y); $a_x$ and $b_y$ are the percentages of edges connecting nodes with values x and y, and $\sigma_a$ and $\sigma_b$ are the standard deviations of $a_x$ and $b_y$ (**Newman, 2003**). The efficiency, in turn, captures how easily the graph can be traversed. For vascular graphs, efficiency can be conceptualized as the average of the inverse of the total resistive distance of the shortest paths of all combinations of vascular junctions, with the hydraulic resistivity serving as the distance between them. The resistivity (**Equation 7**) is summed along the shortest paths, and the inverse of this sum is then averaged across all shortest paths to compute the efficiency (**Equation 8**). Finally, photostimulation-induced change in the efficiency, from its baseline level, is reported.

Resistivity

$$\rho = \frac{8\mu L}{\pi R^4} \tag{7}$$

Efficiency

$$E = \frac{1}{N(N-1)} \sum_{i \neq j \in V} \frac{1}{\rho_{ij}} \tag{8}$$

In computing resistivity, we assumed a fluidic viscosity (μ) of 4 cP which is within the physiological range (*Nader et al., 2019*); L was the length of the capillary; and R was the radius of the capillary. Vessels greater than 10 μm in diameter were excluded from the efficiency calculation as we wanted to examine blood flow through the capillary nexus between arteries and veins where blood flow may be reversible (*Schaffer et al., 2006*). The sum of the resistivities along the least resistive path specified $\rho_{ij}$ in the efficiency calculation: this sum quantifies how hard it is for blood to move through the said microvascular bed. N refers to the number of nodes in the network. The efficiency was calculated as the average of the inverse of the least resistive paths between all pairs of nodes. The efficiency increases with increasing radius on the shortest paths through the network.

### Statistical models

To statistically compare deep learning model performance, we used the Wilcoxon signed-rank test as implemented in *scipy* (1.9.3) (*Virtanen et al., 2020*). Statistics for vascular radius and network metrics were performed in R (4.3.1) using restricted maximum likelihood mixed effects models as implemented in lme4 (1.134; *Bates, et al., 2015*), with post hoc comparisons done using *emmeans* (1.8.8; *Lenth, 2023*). We ran the following linear mixed effects model, separately on dilating and constricting vessels that exhibited radius changes above two standard deviations of the vessel's baseline radius, to examine the effects of photostimulation power on microvascular radius changes in responding vessels:

$$\Delta \text{Radius} \sim \text{Stimulation} + (1|\text{Vessel})$$

Due to the difference in their vessel wall ultrastructure, larger microvessels (above 5 μm in radius) were examined separately from capillaries (radius <5 μm).

For graph metrics, we included nesting of the field of view within each subject as a random effect so as to account for differences in vascular network architecture within an individual. The linear mixed effects models for the graph metrics used were as follows:

$$\text{Assortativity} \sim \text{Stimulation} + (1|\text{Subject}/\text{Field of View})$$

$$\Delta \text{Efficiency} \sim \text{Stimulation} + (1|\text{Subject}/\text{Field of View})$$

In Tables and text, all values have been quoted as mean ± standard deviation, unless otherwise specified.

## Results

Application of our computational pipeline resulted in robust segmentation of the vasculature and neurons from 4D in situ 2PFM images and rendering of the microvasculature as a graph. The vessel-wise and vertex-wise calibers were tracked across stimulation conditions and related to the cortical depth and the distance to the closest YFP-expressing neuron, mapping the network-level vascular responses to ChR2 activation and revealing the coordination of the microvascular responses following neuronal activation.

### Segmentation model results and comparisons

We compared an ensemble of UNETR models, an ensemble of U-Net models, and an ilastik random forest model on a test dataset of nine images (507x507x250 μm each) from six mice. Examples of segmentation masks produced by each of the models are shown in *Figure 3* and *Appendix 1—figure 4*. When evaluating model performance, we paid close attention to the smoothness of the surface of the segmentation masks due to the sensitivity of the centerline extraction algorithms to irregularities in the surface of the masks: smoother vascular segmentation masks resulted in fewer falsely identified vessel branches. Ilastik tended to over-segment vessels, that is the model returned numerous false positives, having a high recall (0.89±0.19) but low precision (0.37±0.33; *Figure 4*, *Supplementary file 3, table 3*). When comparing the UNETR and U-Net models, we focused on the surface-based mean surface distance and HD95 distance metrics. Since we observed no significant differences in these metrics between the two models, we selected the UNETR model as our final model because it produced more consistent segmentations on visual inspection and showed significantly better performance than ilastik on HD95 for both vessel and neuron segmentation (p < 0.05).

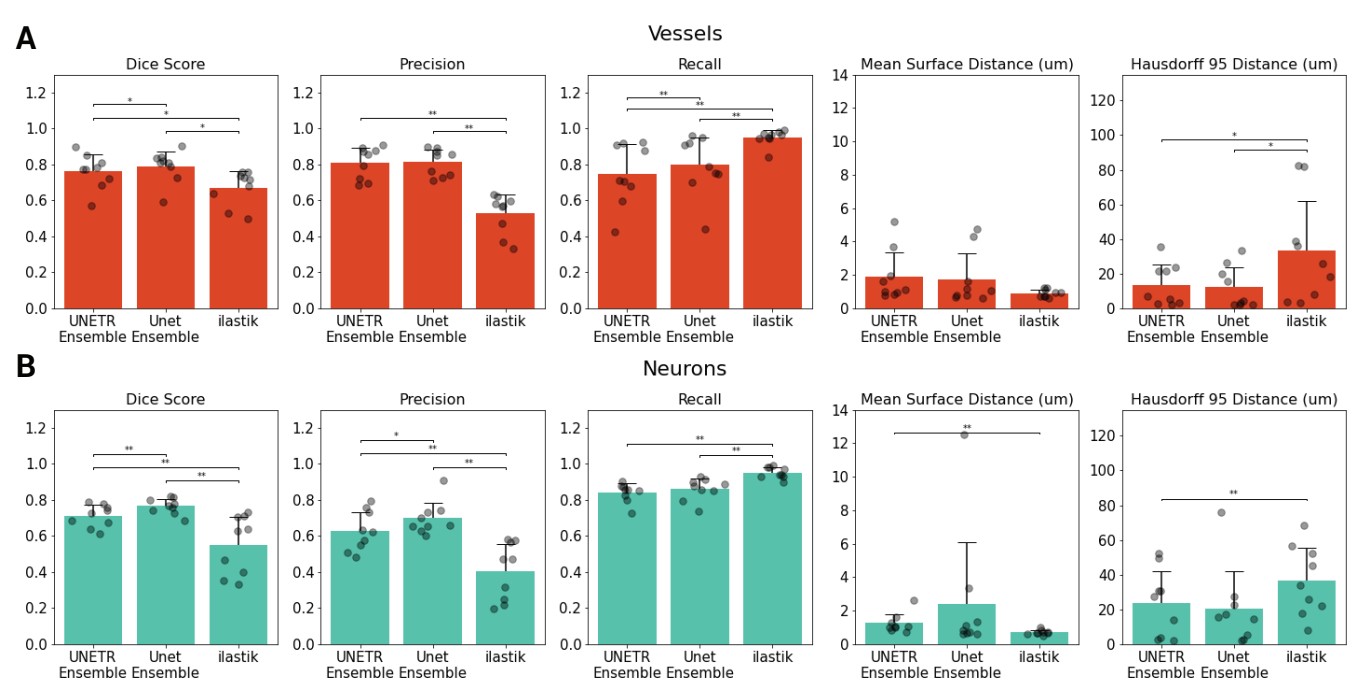

**Figure 3.** Model performance metrics. The Dice, precision, recall, mean surface distance, and HD95 distance for the vascular (**A**) and neuron (**B**) channels. Each model was evaluated on the same test dataset composed of nine images (250 x 507 × 507 μm each) from six mice. A Wilcoxon signed-rank test was used to compare the model's performance on each performance metric for images from the test dataset. * p<0.05, ** p<0.005, and *** p<0.0005. p-values were not adjusted.

## Vessel extraction improvements via image registration

Rigid registration across all time points from the same field of view improved the ability to trace vascular paths from in situ 2PFM data. Firstly, registration decreased the mean squared error (MSE) between acquisitions from 1306±747–0.008±0.003 Signal Units. The number of images acquired per field of view ranged from a total of 2–10 depending on how many repeats were able to be acquired. Registration enabled the computation of the union of segmentation masks from all time points. This increased the number of vessel segments identified in each field of view from 241±174 based on a single time point to 412±281 vessel segments per field of view (507 x 507 × 250 μm, n=107 fields of view). Taking the union of segmentation masks of an image stack across all time points substantially decreased the incidence of gaps in capillaries, likely arising due to 'transient RBC plugs'. The pipeline's ability to reconstruct the cortical vascular network was thus significantly improved by registering data obtained at different time points.

## Validation of pipeline sensitivity to geometric changes

To evaluate the ability of our computational pipeline to detect vessel caliber changes of various magnitudes, we simulated a range of vascular caliber changes and injected various levels of Gaussian noise. Across > 100,000 simulations, the fit of the estimated radius following rescaling against the simulated radius had an $R^2$ value of 0.68. ***Figure 5B*** presents a heatmap of the estimated radius post-scaling vs. simulated radius, across different vertices of vessel centerlines, highlighting the ability of our pipeline to estimate vascular radii accurately. The addition of Gaussian noise revealed the robustness of the pipeline: radius estimates remained stable with increasing noise levels, until the addition of noise with a standard deviation of over 200 SU (with the intensity of the images ranging from 0 to 1023 SU).

Our boundary detection algorithm successfully estimated the radius of precisely specified fluorescent beads. The bead images had a signal-to-noise ratio of 6.79±0.16 (about 35% higher than our in vivo images): to match their SNR to that of in vivo vessel data, following deconvolution, we added Gaussian noise with a standard deviation of 85 SU to the images, bringing the SNR down to 5.05±0.15. The data processing pipeline was kept unaltered except for the bead segmentation,

performed via image thresholding instead of our deep learning model (trained on vessel data). The bead boundary was computed following the same algorithm used on vessel data: that is by the average of the minimum intensity gradients computed along 36 radial spokes emanating from the centerline vertex in the orthogonal plane. To demonstrate an averaging-induced decrease in the uncertainty of the bead radius estimates on a scale that is finer than the nominal resolution of the imaging configuration, we tested four averaging levels in 289 beads. Three of these averaging levels were lower than that used on the vessels, and one matched that used on the vessels (36 spokes per orthogonal plane and a minimum of 10 orthogonal planes per vessel). As the amount of averaging increased, the uncertainty on the diameter of the beads decreased, and our estimate of the bead's diameter converged upon the manufacturer's Coulter counter-based specifications (7.32±0.27 µm), as tabulated below in *Table 1*.

## Vascular morphology and heterogeneity within and among vessels

Segmentation coupled with graph extraction enabled a detailed characterization of the microvascular network properties. The morphological properties of extracted networks are listed in *Table 2*, with the probability densities of the vessel length, baseline vessel radius, mean vessel segment depth, and vessel branch point depth shown in *Appendix 1—figure 5*. On the extracted graphs, the vascular

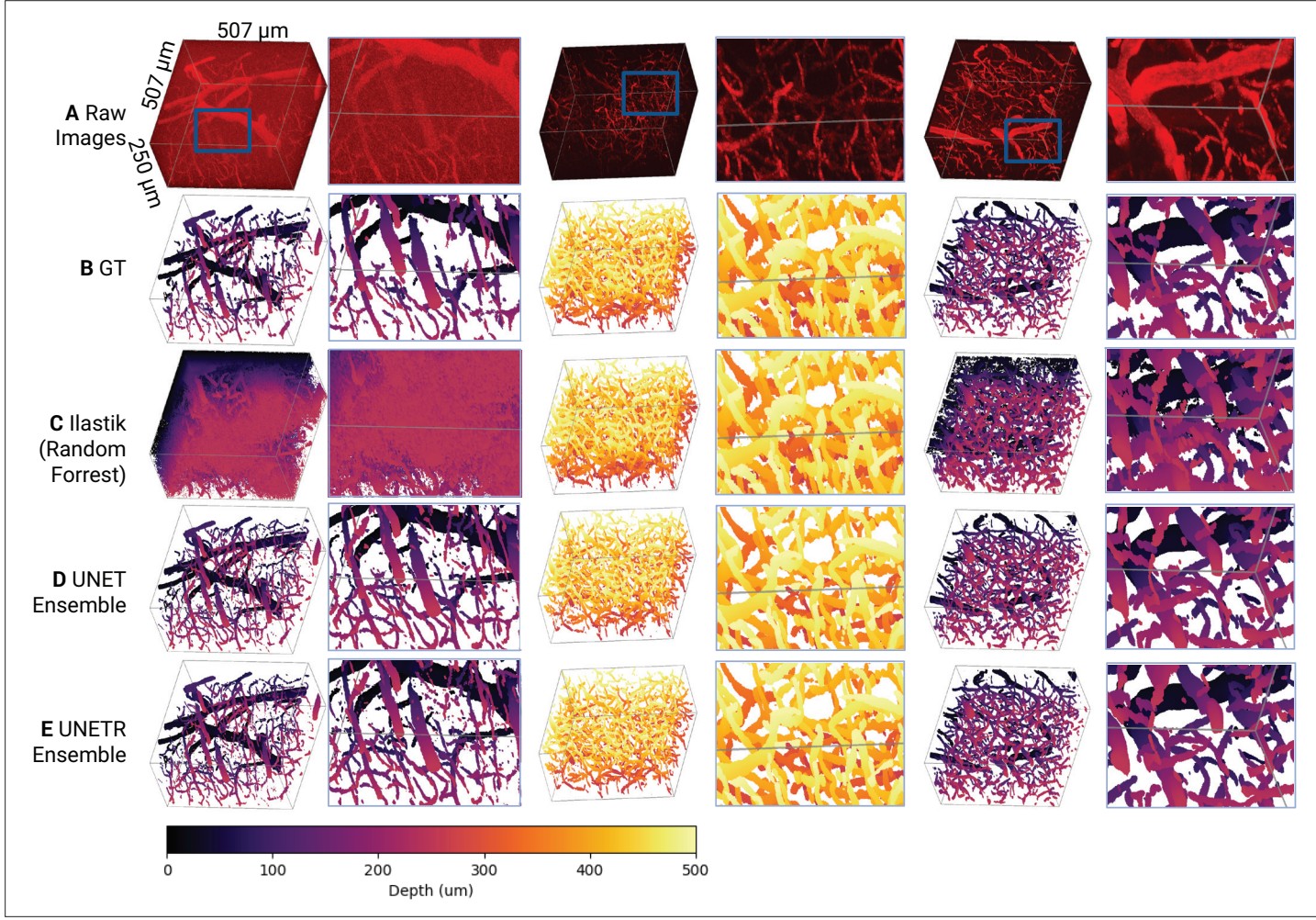

**Figure 4.** Visual model comparison. (**A**) Raw images of the vascular channel with the neuron channel subtracted to facilitate vessel visualization. The first and last stacks in each row span from the cortical surface to 250 µm below the surface, while the middle stack spans from 250 µm below the surface to 500 µm below the surface. All images were from the test dataset, which was unseen during model training. (**B**) Ground truth segmentation masks for the vasculature were generated by a rater who utilized ilastik-assisted manual segmentation. (**C**) Ilastik predictions generated via a random forest model. (**D**) Binary segmentation masks generated by an ensemble of 3D UNet models. (**E**) Binary segmentation masks generated by an ensemble of 3D UNETR models.

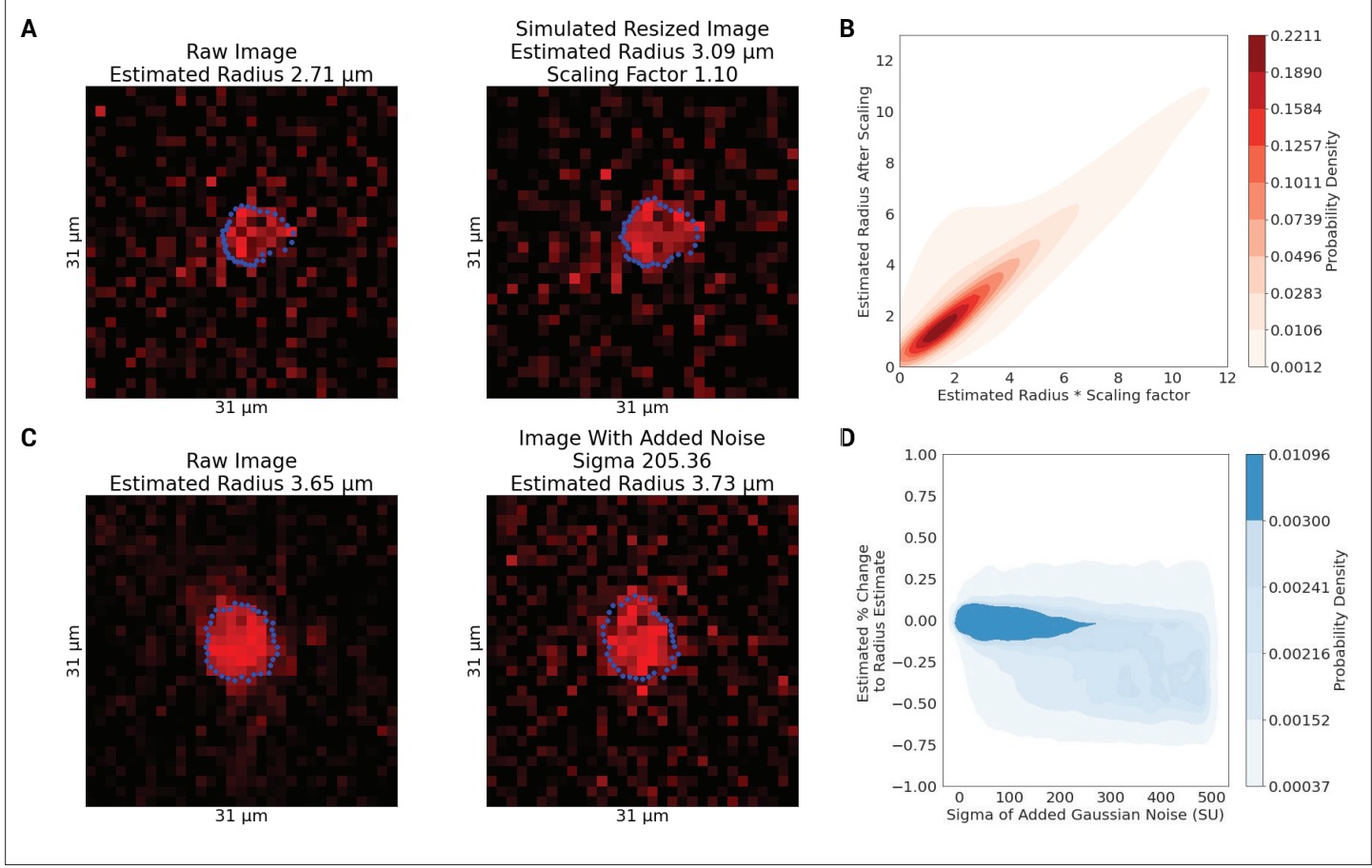

**Figure 5.** Estimation of simulated radii changes. (**A**) An image in the plane orthogonal to the local tangent to a capillary with the detected boundary (in blue) and with the estimated radius of 2.28 µm. On the right, this image was resized (upsampling, via bicubic interpolation, by 1.10 times) to simulate dilation. (**B**) The plot shows correspondence between the estimated radius following scaling and the simulated level of scaling. (**C**) An image in the plane orthogonal to the local tangent of a capillary with the detected boundary (in blue) and with the estimated radius of 3.65 µm. On the right, Gaussian noise with a sigma of 205.36 SU was added to the image. (**D**) The estimated % change in the vessel's radius after the addition of varying levels of Gaussian noise, demonstrating the robustness of the radius estimated to noise.

radius and distance to labeled neurons were sampled every 1–1.73 µm, enabling detailed analysis of the relationship between the vessel radius change and the proximity to the YFP-expressing neurons. The radius was tracked across different time points, permitting the analysis of the stimulation-induced change in the vascular caliber. To highlight the ability of the pipeline to detect vessels that significantly change their radius after stimulation, *Figure 6A* shows the standard deviation of the average radii on each vessel segment during baseline frames for three mice. There was a large difference in this standard deviation across various blood vessels, showcasing the model's ability to reveal baseline variations within each subject. We examined the average change in the vascular radius of each vessel segment after vs. before photostimulation (*Figure 6B*), with even finer spatial patterns detected by analyzing the vertex-wise radius changes (*Figure 6C*). The vascular diameter changes were related

**Table 1.** Bead diameter estimates.

| Number of orthogonal planes | Number of spokes per plane | Mean diameter estimate (µm) |
| --- | --- | --- |
| 1 | 3 | 7.54±0.68 |
| 2 | 4 | 7.44±0.51 |
| 4 | 12 | 7.34±0.38 |
| 10 | 36 | 7.34±0.32 |

**Table 2.** S1FL vascular network morphological properties.

| Metric | Mean ±SD | N=17 Mice (9 M/8 F) |
|---|---|---|
| Number of individual vessels per volume | 368±239 | 32 FOVs |
| Vessel density | 5705±3705 mm$^{-3}$ | 32 FOVs |
| Number of vascular junctions per volume | 207±154 | 32 FOVs |
| Vascular junction density | 3215±2385 mm$^{-3}$ | 32 FOVs |
| Number of terminal vessels per volume | 128±52 | 32 FOVs |
| Individual vessel length | 70.7±61.1 µm | 12555 vessel segments |
| Cumulative vessel length density | 0.40±0.22 m/mm$^3$ | 32 FOVs |
| Baseline vessel radius | 2.19±1.66 µm Range: 0.66–15.88 µm | 12555 vessel segments |
| Baseline intra-vessel radius standard deviation | 0.53±0.47 µm | 12555 vessel segments |
| Baseline vascular volume density | 0.010±0.007 mm$^3$/mm$^3$ | 32 FOVs |
| Number of pyramidal neurons per volume | 313±202 neuronal somas | 32 FOVs |
| Pyramidal neuron density | 4872±3145 neuronal somas/mm$^3$ | 32 FOVs |

to the distance from the vessel's surface to the closest labeled pyramidal neuron at each vertex of the centerline (*Figure 6D*). The vertex-wise radii estimation allowed the assessment of the variations in radii changes within individual blood vessels (*Figure 7*). Notably, capillary radius varied along the vessel length across the baseline frames by 24±28% of the mean resting radius. Consequently, point measurements in vessel calibers - that are widely reported in the literature - do not permit accurate estimation of the microvessel volume changes. Together, the within- and across-vessel radii

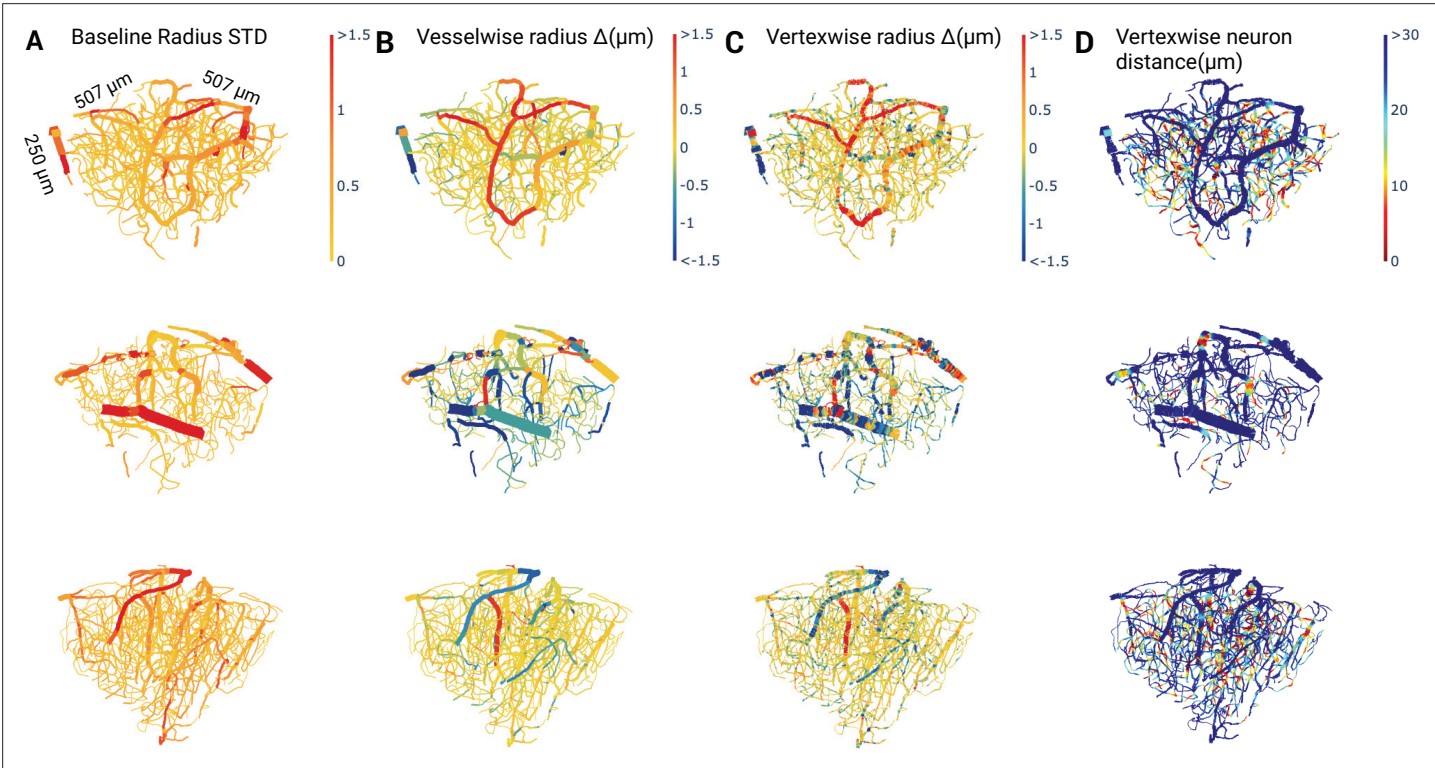

**Figure 6.** Vascular graph examples. (**A**) Baseline variability in vessel diameter estimated by the standard deviation of each vessel's mean radius across baseline time frames. (**B**) Mean change in the vessel radius induced by optogenetic stimulation. (**C**) Mean change in the vertexwise radius, allowing the visualization of heterogeneity of radius changes within each vessel. (**D**) Distance from each vertex to the closest pyramidal neuron. Each row corresponds to the vascular graph of a different mouse.

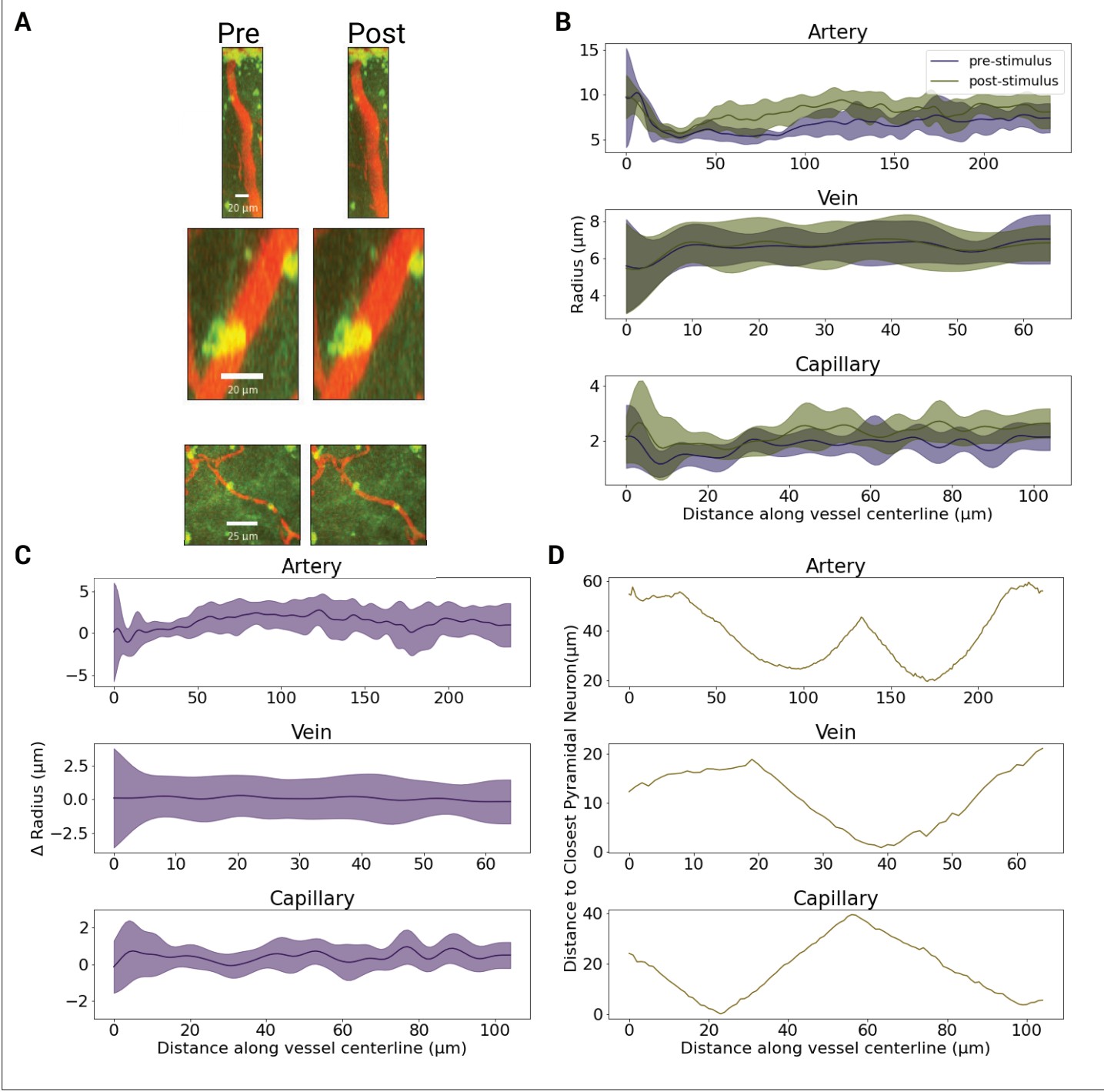

**Figure 7.** Vertex-wise radii along vessel lengths of a sample artery, capillary, and venule at baseline vs. post-stimulation. (**A**) MIP of an artery, vein, and capillary segments before (left) and after (right) optogenetic stimulation with 458 nm light at 1.1 mW/mm². The artery and capillary dilated by 1.33±0.86 µm and 0.42±0.39 µm, respectively (for both p<1e-4, Mann-Whitney U test), whereas there was no significant change in the venular caliber upon photostimulation (p=0.22, Mann-Whitney U test). (**B**) Estimates of the vertex-wise radius obtained along each of the three vessels' centrelines, before and after stimulation. (**C**) Vertex-wise radii changes in response to optogenetic stimulation. (**D**). The vertex-wise distance from the vascular surface to the closest YFP-expressing neuron.

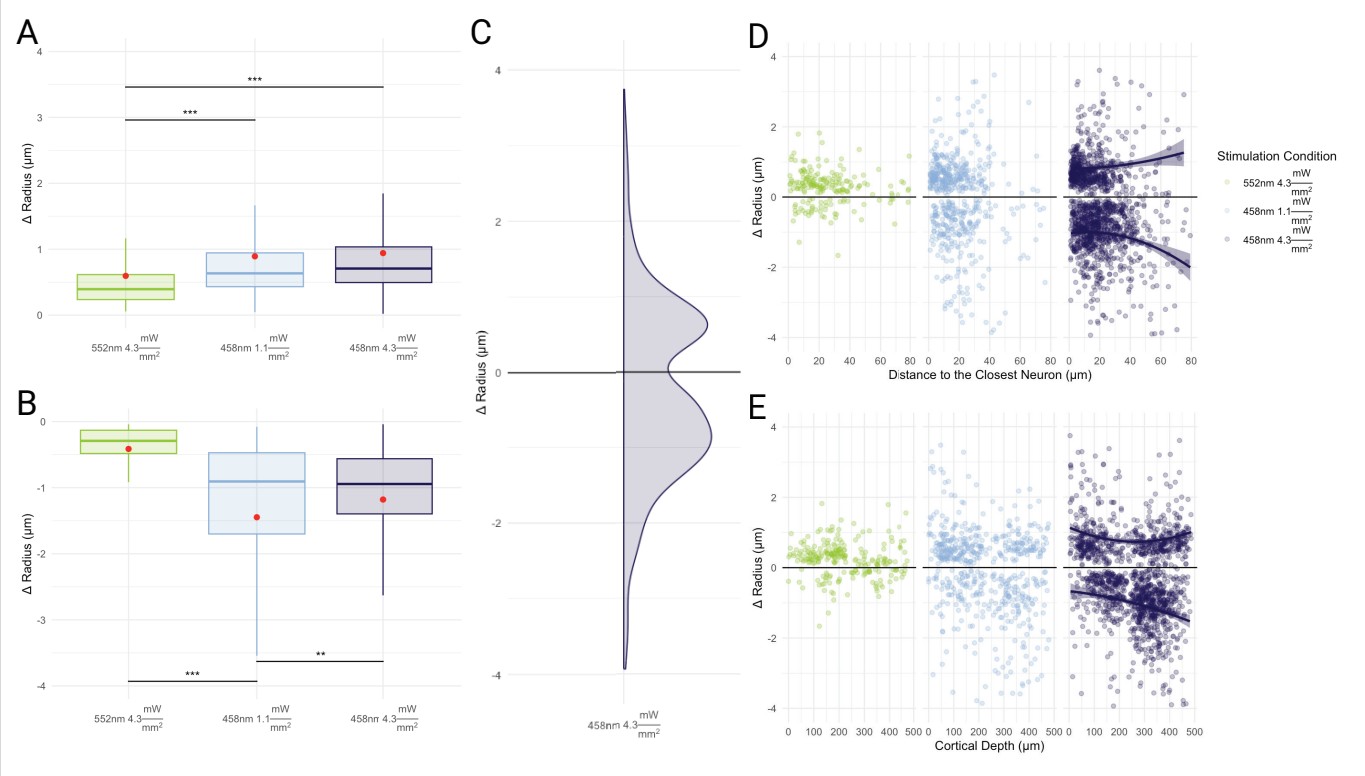

**Figure 8.** Optogenetic activation-induced changes in vessel-wise microvascular radii. Capillary responses included both dilatations, shown in (A), and constrictions, shown in (B), with changes in the magnitude of the capillary response with increased photostimulation power. * p<0.05, ** p<0.005, and *** p<0.0005. p-values were not adjusted. (C) Probability density function of constrictions and dilations for the 4.3 mW/mm² photostimulation. (D) Changes to capillary radii are displayed in relation to the closest pyramidal neurons. The proportion of vessels constricting increased with the higher intensity of blue light stimulation, and constrictions tended to occur further away from pyramidal neurons than did dilations. (E) Mean cortical depth of responding capillaries showed a tendency for dilators to be closer to the surface and for constrictors to be deeper in the tissue.

estimations illustrate the pipeline's ability to capture spatial variations in the vascular reactivity and relate it to other morphological features (e.g. the distance to the closest labeled neuron).

## Vascular reactivity to optogenetic stimulation

The ability of the pipeline to reveal novel spatial relationships between the vascular network reactivity and labeled neurons was demonstrated by examining the relationship between photostimulation-induced microvascular radii responses and (1) the closest YFP-labeled pyramidal neurons within 80 μm, and (2) the cortical depth, at the vertex-wise level and across different photostimulations (*Figure 8*). Vessels were coarsely segregated into small (average radius <5 μm) and large (average radius >5 μm) vessels, as we expected them to respond differently due to their differential wall-associated cell composition (*Hartmann et al., 2021*; *Bisht et al., 2021*; *Wu et al., 2022*; *Kirabali et al., 2019*; *Ren et al., 2021*; *Steinman et al., 2017*; *Berthiaume et al., 2018*; *Katz et al., 2023*; *Drouin-Ouellet et al., 2015*). Only vessels longer than 20 μm (i.e. vessels whose radius was computed by averaging over many cross-sectional planes) that significantly responded following optogenetic stimulation were analyzed: a vessel was deemed a responder if its radius changed by more than twice the baseline standard deviation in the vessel's radius. The morphometric properties of the responders, under different stimulation conditions, are listed in *Table 3*. The average magnitude of significant microvascular radius changes across all stimulation conditions was 1.04±1.11 μm (62 ± 47%). The variability in the radius change within the vessel was higher in the dilating vessels, 0.77±0.61 μm (66 ± 72%) than in the constricting vessels, 0.69±0.49 μm (46 ± 18%), for 458 nm, 4.3 $\frac{mW}{mm^2}$ photostimulation (p<1e-4; with no statistically significant changes detected for either 458 nm, 1.1 $\frac{mW}{mm^2}$ photostimulation or 552 nm, 4.3 $\frac{mW}{mm^2}$ photostimulation). Excluding vessels that did not change their radius by over twice the baseline standard deviation removed almost all large vessels from this analysis. (Notwithstanding,

**Table 3.** Details of responder ($\Delta R > 2 * \sigma R_{baseline}$) vessels.

| Stimulation condition | Total number of vessel estimates | Average minimum distance to the closest neuron (μm) | Number of dilators | Minimum distance from dilators to the closest neuron (μm) | Average vessel depth of dilators (μm) | Diameter change (μm) | Number of constrictors | Minimum distance from constrictors to the closest neuron (μm) | Average vessel depth of constrictors (μm) | Diameter change (μm) |
|---|---|---|---|---|---|---|---|---|---|---|
| All vessels | | | Dilators | | | | Constrictors | | | |
| Capillaries | | | | | | | | | | |
| 552 nm 4.3 $\frac{mW}{mm^2}$ | 5036 | 21.2±16.2 | 144 (2.9%) | 25.5±19.0 | 186±114 | 0.58±0.92 | 49 (1.0%) | 26.5±19.5 | 247±122 | −0.37±0.30 |
| 458 nm 1.1 $\frac{mW}{mm^2}$ | 10136 | 18.7±14.5 | 317 (3.1%) | 16.8±13.5 | 196±138 | 0.90±0.93 | 255 (2.5%) | 22.7±16.3 | 254±126 | −1.39±1.51 |
| 458 nm 4.3 $\frac{mW}{mm^2}$ | 12537 | 20.6±15.4 | 575 (4.6%) | 16.1±14.3 | 237±146 | 0.90±0.77 | 874 (7.0%) | 21.9±14.6 | 274±103 | −1.19±1.13 |
| Large vessels | | | | | | | | | | |
| 552 nm 4.3 $\frac{mW}{mm^2}$ | 225 | 43.1±19.5 | 1 (0.4%) | 75.4 | 82 | 13.98 | 0 (0%) | NA | NA | NA |
| 458 nm 1.1 $\frac{mW}{mm^2}$ | 545 | 38.4±19.5 | 1 (0.2%) | 26.1 | 402 | 1.97 | 1 (0.2%) | 19.0 | 179 | −3.65 |
| 458 nm 4.3 $\frac{mW}{mm^2}$ | 569 | 38.4±20.1 | 2 (0.35%) | 53.1±6.3 | 84±34 | 2.47±2.93 | 6 (1.1%) | 43.1±16.3 | 290±125 | −6.07±2.45 |

*Appendix 1—figure 5* depicts unfiltered large vessel constrictions and dilations). We ran mixed effects models (at the vessel level) separately on constricting and dilating vessels to investigate the effect of optogenetic stimulation power on the vessel radius changes. Each of the models was run separately on small and large vessels.

## Vessels further away from labeled neurons constrict while those closer to the activated neurons dilate

We examined the relationship between vascular radius changes and the distance to the closest labeled pyramidal neuron, as microvascular response is thought to result from neuronal activation-elicited generation of vasoactive molecules that diffuse to the neighboring vessels. For the control condition (552 nm, 4.3 $\frac{mW}{mm^2}$ photostimulation), 2.9% of small capillaries dilated while 1.0% of small capillaries constricted; in larger vessels, barely any responded (0.4% dilated). For this control condition, there was no significant difference in the distance from constrictors or dilators to the closest pyramidal neuron. For the 458 nm photostimulation, capillary constrictors were on average farther away than were dilators from the labeled pyramidal neuron: dilations occurred 16.8±13.5 μm away from labeled neurons while constrictions occurred 22.7±16.3 μm for 1.1 $\frac{mW}{mm^2}$ photostimulation (p=1.5e-3) whereas the 4.3 $\frac{mW}{mm^2}$ photostimulation had dilations occur 16.1±14.3 μm away and 21.9±14.6 μm for constrictors (p<1e-4). There was no significant shift between the distance from vessels to neurons for the 1.1 $\frac{mW}{mm^2}$ and 4.3 $\frac{mW}{mm^2}$ stimulations with 458 nm light. Dilations in capillaries following 458 nm photostimulation were larger than those following the 552 nm control photostimulation: 0.90±0.93 μm dilatations occurred with 1.1 $\frac{mW}{mm^2}$ and 0.90±0.78 μm with 4.3 $\frac{mW}{mm^2}$ at 458 nm; vs. 0.58±0.92 μm with 4.3 $\frac{mW}{mm^2}$ at 552 nm (p<1e-4). For constrictions, 458 nm photostimulations led to −1.39±1.51 μm radius changes with 1.1 $\frac{mW}{mm^2}$ and −1.20±1.13 μm radius changes with 4.3 $\frac{mW}{mm^2}$ (p=4.4e-3), whereas 552 nm photostimulation induced −0.37±0.30 μm radius changes with 4.3 $\frac{mW}{mm^2}$ of power, which was smaller than the 458 nm induced responses (p=0.02).

## Vascular radius changes at increasing cortical depths

Vascular responses were next segregated by the cortical depth of the vessel (i.e. the average vessel distance from the cortical surface; *DeFelipe et al., 2002*). Dilators tended to be located closer to the cortical surface across all stimulation conditions. Constricting vessels were located at an average 58±187 μm deeper than dilators for 458 nm stimulation at 1.1 $\frac{mW}{mm^2}$ (p=0.02), and 37±179 μm deeper for 458 nm photostimulation at 4.3 $\frac{mW}{mm^2}$ (p<1e-4; with no change in the mean depth of either constricting or dilating vessels with changes in the photostimulation power).

Vascular network coordination following optogenetic stimulation

We examined the coordination of changes in the microvascular network as a whole via assortativity of radius changes and network efficiency changes. The vessel responses were observed to be assortative, that is capillaries mirrored the responses in their neighbors. The increases in stimulation power were accompanied by increases in the assortativity of capillary responses: increasing stimulation level resulted in heightened coordination between adjacent capillaries (*Figure 9*).

The efficiency increased only at the strongest blue photostimulation, that is only at this level of stimulation did the resistivity along the average of all of the shortest paths between junctions in the vascular network decrease, resulting in attenuated resistance to flow through the network. The distribution of changes to the efficiency was highly skewed (with a coefficient of skewness of $-1.06$ for green illumination, $2.92$ for lower intensity blue photostimulation, and $4.87$ for higher intensity blue photostimulation). The median increase in the efficiency induced by the higher intensity blue photostimulation, of 4% (IQR: $-8\%$ to 38%), was significantly higher than the median $-6\%$ (IQR=$-9$–4%) efficiency change following the control green illumination.

## Discussion

Recent studies have demonstrated temporal propagation and coordination in cerebrovascular responses to neuronal activation, whereby arteries dilated after capillary exposure to increased potassium ion concentration (*Dabertrand et al., 2021*); opposing geometric changes have been reported by some studies in capillaries connected by intercapillary tunneling nanotubes (*Alarcon-Martinez et al., 2020*). However, how the effects of these and other mechanisms influence in situ reactivity of the 3D brain capillary network remains unknown. Here, we developed a pipeline for extracting graphs of brain microvascular networks from in situ 2PFM and examining coordination within and across capillaries. Capillary networks and their geometrical changes were imaged via 2PFM during periods of baseline alternated with photostimulation of ChR2 in pyramidal neurons of transgenic mice, and the microvascular mesh was evaluated every 1–1.73 µm. The vascular morphology was then analyzed vertex- and vessel-wise across the entire network. All vessels exhibited significant heterogeneity in caliber changes along their length. Neuronal activation induced both dilations and constrictions of vessels, and the incidence of constrictions increased with increasing cortical depth. As the stimulation power increased, the tendency for vessels to change their radius by an amount similar to their neighbors increased. Only the highest photostimulation intensity elicited an increase in the network efficiency. Our findings reveal an intricate level of coordination among brain microvessels and provide a computational analysis platform for interrogating a host of hypotheses on cerebral microvascular reactivity.

### Vascular segmentation and network extraction

Intensity thresholding-based image processing pipelines have been used to examine vascular networks and quantify vascular morphology, but they have not gained widespread use due to difficulties in adapting them to highly heterogeneous levels of noise across samples (*Steinman et al., 2017*; *Tsai et al., 2009*; *Rennie et al., 2011*; *Lindvere et al., 2013*). Deep learning-based methods for analyzing vascular morphology from 3D microscopy images have become prevalent as they provide robust segmentation results over a wide range of signal-to-noise ratios. Recent work has demonstrated steady improvements in segmentation models' performance with respect to similarity-based metrics (i.e. Dice scores; *Damseh et al., 2018*; *Goodarzi Ardakani et al., 2022*; *Livne et al., 2019*; *Mookiah et al., 2021*; *Tetteh et al., 2020*; *Poon et al., 2023*), although surface-based metrics may be better predictors of how amenable segmentation outputs will be to subsequent morphological analysis of the microvascular network. Our final UNETR model was selected based on a combination of performance metrics including mean surface distance, Hausdorff 95% distance, and rater evaluation, to maximize the smoothness of the surface of generated segmentation masks and reduce false positive branch points during centerline extraction; thereby leading to higher fidelity rendering of microvascular networks. Graph generation was greatly facilitated by computing the union of the vascular segmentation masks across all time points as it enabled tracing of capillaries that had stalls at the individual time points (since the accompanying loss of the fluorescent label otherwise resulted in graph discontinuities).

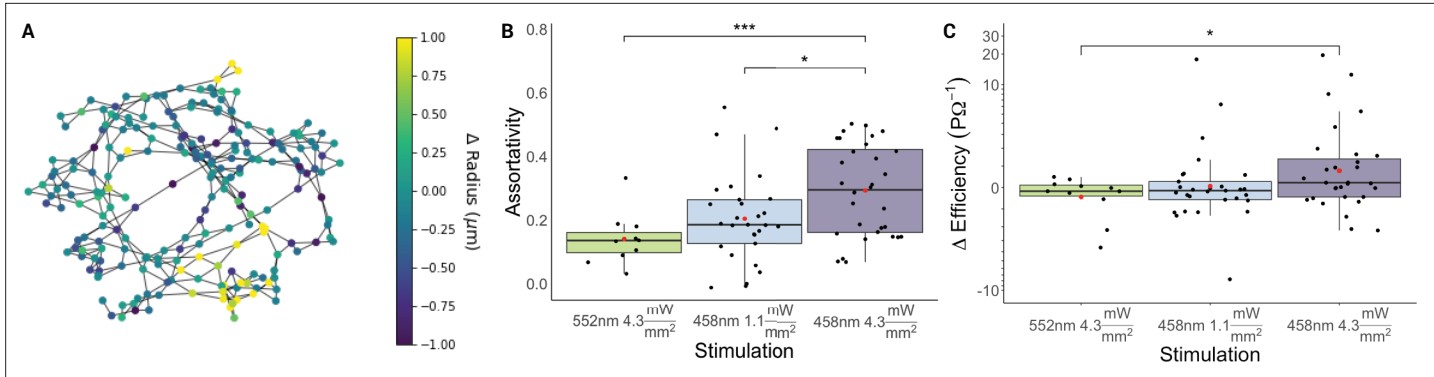

**Figure 9.** Microvascular network coordination following optogenetic stimulation. (**A**) Graph representation of a vascular network of 425 vascular segments from a single image stack. Vessel segments are depicted as nodes of the graph; vascular segments that are joined at junctions are connected by edges. Nodes are colored by the change in the mean vessel-wise radius following photostimulation with 458 nm light at 4.3 mW/mm². (**B**) Assortativity of photostimulation-induced changes in mean capillary radius increased with increasing photostimulation power. (**C**) Photostimulation-induced changes in the efficiency of the capillary network. The capillary network efficiency changed by a median –0.16 PΩ⁻¹ (IQR: –0.39–0.10 PΩ⁻¹) in response to green light; –0.14 PΩ⁻¹ (IQR: –0.55–0.27 PΩ⁻¹) in response to lower intensity blue light; and 0.22 PΩ⁻¹ (IQR = –0.43;1.47 PΩ⁻¹) in response to higher intensity blue light. There was a significant increase (p=0.03) in the capillary network efficiency post 458 nm light at 4.3 mW/mm², when compared to that following the control green illumination. The measurements came from 72 paired acquisitions of 32 image stacks acquired in 17 mice (9 M/8 F). * p<0.05, ** p<0.005, and *** p<0.0005. p-values were not adjusted.

We tested the ability of the pipeline to detect changes to simulated changes to images and the pipeline's sensitivity to perturbations to extracted vascular morphology. Using resizing, we confirmed that the boundary detection algorithm was able to detect prescribed changes (*Figure 5*). The average radius estimates for vessels were then shown to vary by 0.64 ± 3.44% upon changes in centerline position, demonstrating that the radius estimates have a low sensitivity to small (2–3 µm) perturbations in centerline placement. Visual examples of vessels repeatedly dilating or constricting are shown in *Appendix 1—figures 7 and 8*.

Network morphological properties at baseline were in line with prior work. The vascular length density measured from fixed tissue ranged from 0.44 to 1.10 m/mm³ (*Tsai et al., 2009*; *Todorov et al., 2020*; *Boero et al., 1999*; *Lugo-Hernandez et al., 2017*; *Zhang et al., 2018a*; *Miettinen et al., 2021*). Our reported vascular density in the forelimb region of the primary somatosensory cortex was 0.40±0.22 m/mm³, with the low end of the range value expected due to fluorescence absorption by hemoglobin in the large pial vessels leading to signal dropout (or shadowing) in the underlying tissue. Our reported average capillary radius of 2.19±1.66 µm was also in line with other studies, where the mean capillary radius ranged from 1.75 to 2.2 µm as measured with confocal microscopy or 2PFM (*Hall et al., 2014*; *Tsai et al., 2009*).

## Next in situ changes to vessel calibers upon neuronal activation

Caliber changes at individual vertices along vessel centerlines exhibited significant heterogeneity. Such heterogeneity is expected due to non-uniformly distributed alpha smooth muscle actin-containing cells along vessel walls, as well as differential activations leading to heterogeneous metabolic demand within the tissue (*Iadecola, 2017*; *Hartmann et al., 2021*; *Kovacs-Oller et al., 2020*; *Abdelazim et al., 2022*; *Quelhas et al., 2020*; *Wang et al., 2015*). Many previous studies assumed vessel caliber changes to be uniform, compromising the accuracy of the estimates.

As expected, the control 552 nm stimulation led to minimal changes in vessel calibers. To probe for off-target effects, non-transgenic mice were also tested with the same optical setup and photostimulation, with no changes to vascular diameters observed at any of the photostimulation powers utilized (*Appendix 1—figure 9*). In transgenic mice, we detected an average capillary dilation in significantly responding vessels of 70 ± 83% with low-intensity 458 nm stimulation, and 67 ± 61% with higher intensity 458 nm stimulation. Across photostimulation conditions, the capillary dilations ranged from 2% to 805%. These caliber changes were higher than those previously reported, which varied from 2% to 20% depending on the capillary branch order (*Hartmann et al., 2021*; *Hall et al., 2014*; *O'Herron et al., 2022*; *Del Franco et al., 2022*; *Stefanovic et al., 2008*). Far less data are available

on constrictions. In the current work, constrictions averaged 47 ± 20% for lower-intensity blue light stimulation and 47 ± 17% for higher-intensity blue light stimulation, with a constriction range of 5% to 97% of the baseline radius. These are higher than the previously reported constrictions of 20% (*Hartmann et al., 2021*; *O'Herron et al., 2022*), likely due to our identifying as responding vessels only those whose caliber changed by at least twice their baseline caliber variation. It is also worth noting that vessels' response directions were consistent on repeated trials. Of the vessels whose radius change exceeded twice the baseline variability across time, 31.7% dilated on some trials while constricting on others; 41.1% dilated on each trial; and 27.2% constricted on each trial. (Note that some trials use 1.1 vs 4.3 mW/mm² and some have opposite scanning directions).

458 nm photostimulation resulted in a mix of constrictions and dilations with 44.1% of significantly responding vessels within 10 µm of a labelled pyramidal neuron constricting and 55.1% dilating, while 53.3% of vessels further than 30 µm constricted and 46.7% dilated. The cutoff distances from the closest labeled neuron were based on estimates of cerebral metabolic rate of oxygen consumption that showed a steep gradient in oxygen consumption with distance from arteries, CMRO2 being halved by 30 µm away (*Mächler et al., 2022*). The stronger blue light stimulation led to an increased rate of constrictions, double that of the low-powered blue light stimulation. For larger vessels, both 458 nm stimulation powers led to a similar dilation level that diminished with increasing distance from labeled pyramidal neurons. This tendency for vessels close to neurons to dilate and further away ones to constrict would be expected in flow redirection into regions of high level of neuronal activity. Stimulation power dependence in blood flow changes has previously been reported in optogenetic mouse models with diffuse stimulation via LED probes, and following transcranial alternating current stimulation (*Lee et al., 2021*; *Turner et al., 2021*). However, neither of the previously employed methods was able to discern the spatial relationship between the vascular caliber changes, or relate these changes to the distribution of the stimulated neurons.

As the blue light stimulation power increased, the mean depth of both constricting and dilating vessels increased, likely resulting from higher intensity light reaching pyramidal neurons deeper in the tissue (*Johnson et al., 2021*; *Al Juboori et al., 2013*). The blue light would be expected to excite a lower number of neurons farther from the cortical surface at lower powers. Our results underscore that the hemodynamic response following targeted neuronal activation is not uniformly distributed across the microvascular network: accurate neurovascular coupling assessment thus requires network-based analysis.

## Vascular network reactivity

To study the microvascular network response as a whole, we examined the assortativity between capillary radius changes and network efficiency changes following optogenetic stimulation. These two graph theory metrics were selected as they both leverage the knowledge of the vascular network structure. Assortativity sheds light on how the vascular network coordinates its responses, while efficiency provides insight into the extent to which those changes facilitate flow through the network. The assortativity revealed that as the stimulation power increased, the tendency of vessels to match their changes to those of their neighbors increased. Previously characterized assortative mechanisms include endothelial cell cation conduction via Kir2.1 channels to synchronize vascular responses (*Dabertrand et al., 2021*), and spatial adjacency of pericytes on in vitro retinal preparation leading to assortative changes in neighboring capillaries (*Kovacs-Oller et al., 2020*). Disassortative (causing opposite changes) mechanisms of capillary coordination have also previously been observed in situ and may result from intercapillary nanotubes' signaling causing connected pericytes to undergo opposing changes (*Alarcon-Martinez et al., 2020*). While not ruling out the presence of disassortative control mechanisms, our results suggest that assortative mechanisms dominate capillary responses to neuronal activation in the somatosensory cortex.

The network efficiency here can be thought of as paralleling mean transit time, i.e., the time it takes blood to traverse the capillary network from the arteries to the veins. In situ studies of mean transit time have revealed a high heterogeneity of plasma traversal of the capillary bed during stimulation, with stimulation reducing plasma transit times by 11% to 20% from its resting levels (*Stefanovic et al., 2008*; *Gutiérrez-Jiménez et al., 2016*), and simulations suggesting that capillary network geometry and locations of caliber changes exert a substantial influence on these responses (*Lücker et al., 2018*). The efficiency of the vascular network here increased significantly only with the strongest 458 nm

stimulation. Small dilatations may thus not increase flow in the cortex. The differences in efficiency are likely due to the patterns of localized dilations and constrictions within the vascular network. Efficiency calculations are sensitive to bottlenecks when traversing meshes and certain locations constricting or dilating can have profound impacts on the shortest paths between nodes and the path's resistivity. The highest-powered 458 nm stimulation increasing efficiency may have resulted from increased assortativity causing dilation in key locations within the microvascular network, leading to a significant reduction in shortest path resistivity.

## Comparison with commercial and open-source vascular analysis pipelines

To compare our results with those achievable on these data with other pipelines for segmentation and graph network extraction, we compared segmentation results qualitatively with Imaris version 9.2.1 (Bitplane) and vascular graph extraction with VesselVio (*Bumgarner and Nelson, 2022*). For the Imaris comparison, three small volumes were annotated by hand to label vessels. Example slices of the segmentation results are shown in *Appendix 1—figure 10*. Imaris tended to either over- or under-segment vessels, disregard fine details of the vascular boundaries, and produce jagged edges in the vascular segmentation masks. In addition to these issues with segmentation mask quality, manual segmentation of a single volume took days for a rater to annotate. To compare to VesselVio, binary segmentation masks (one before and one after photostimulation) generated with our deep learning models were loaded into VesselVio for graph extraction, as VesselVio does not have its own method for generating segmentation masks. This also facilitates a direct comparison of the benefits of our graph extraction pipeline to VesselVio. Visualizations of the two graphs are shown in *Appendix 1—figure 11*. Vesselvio produced many hairs at both time points, and the total number of segments varied considerably between the two sequential stacks: while the baseline scan resulted in 546 vessel segments, the second scan had 642 vessel segments. These discrepancies are difficult to resolve in post-processing and preclude a direct comparison of individual vessel segments across time. As the segmentation masks we used in graph extraction derive from the union of multiple time points, we could better trace the vasculature and identify more connections in our extracted graph. Furthermore, VesselVio relies on the distance transform of the user-supplied segmentation mask to estimate vascular radii; consequently, these estimates are highly susceptible to variations in the input segmentation masks. We repeatedly saw slight variations between boundary placements of all of the models we utilized (ilastik, UNet, and UNETR) and those produced by raters. Our pipeline mitigates this segmentation method bias by using intensity gradient-based boundary detection from centerlines in the image (as opposed to using the distance transform of the segmentation mask, as in VesselVio).

## Pipeline limitations and adaptability

The segmentation model was trained only on vascular and neuronal labels, limiting its generalizability to segmenting alternative cells in the current state. However, it can easily be fine-tuned or retrained to label other brain cells (e.g. pericytes, astrocytes, or endothelial cells). Our vascular segmentation model generalized well to C57BL/6J mouse and Fischer rat data, as well as to Thy1-ChR2 light-sheet fluorescence microscopy images gathered on an UltraMicroscope Blaze lightsheet fluorescence microscope (Miltenyi Biotech) (*Appendix 1—figures 12 and 13* and *Supplementary file 3, table 3*). However, the segmentation model performed poorly when significant bleeding occurred in the cranial window, compromising the vascular contrast. Our imaging protocol, in turn, was challenged by the desire to resolve individual vessel responses yet capture the entire network within the span of the microvascular response to stimulation: we prioritized network assessment and thereby compromised our temporal sampling (every 42 s), so that our ensuing classification of vessels as dilators or constrictors was based on their caliber at this, rather delayed timepoint. Accordingly, we are unable to comment on finer temporal scale network behavior or the kinetics of the microvascular network response; but the present analysis pipeline can readily be applied to 2PFM data obtained with finer temporal (e.g. via a Piezo objective positioner) or spatial resolution, and/or different size fields of view. The temporal evolution of the response in individual vessels, however, has been reported on using line scanning acquisitions to measure red blood cell velocity and flux and in some cases vessels (*Hartmann et al., 2021*; *Adams et al., 2018*; *O'Herron et al., 2022*; *Mester et al., 2019*; *Kleinfeld et al., 1998*; *Stefanovic et al., 2008*). It is worth noting that the cases where vascular responses

are drawn out following optogenetic stimulation use raster scanning over small regions of interest, and that optogenetic stimulations utilizing fiber optic probes shining light over large areas led to fast vascular responses. Our study utilized raster scanning over small regions of interest. Nevertheless, long-drawn-out vascular responses following optogenetic stimulation remain controversial and still need further study at higher temporal sampling, which our pipeline can readily adapt to, to be demonstrated conclusively. Additionally, alternative definitions of responding vessels may be useful depending on the end goal of a study (e.g. selecting a threshold for the radius change based on a percentage change from the baseline level: *Appendix 1—figure 14* for capillary changes above 10% of the baseline radius). Finally, microvascular networks in different brain areas may show distinct spatiotemporal profiles of response to neuronal activation. Future work is required to test the generalizability of present findings across different brain regions.

## Conclusion

We developed a novel deep learning-based computational pipeline for analysis of a time series of 3D 2PFM images and investigation of spatial patterns in microvascular network reactivity to neuronal activation. The microvascular network was represented as a graph, allowing for the evaluation of network geometry changes over time. We tracked the size of blood vessels throughout the network and related vessel radius changes to the distance from the stimulated neurons and the cortical depth. Neuronal activation induced both dilatations and constrictions of capillaries, and the magnitude of these responses increased with increased photostimulation levels while showing significant heterogeneity within and between vessels. In the analysis presented, vertex-wise measurements were aggregated for vessel-wise analysis, resulting in highly robust estimates of vessels' calibers and allowing ready comparisons to literature. Notwithstanding, the pipeline also affords vertex-wise analysis and thus registration of microvascular reactivity with other local morphological features, at an unprecedented spatial scale. With increasing distance of the vessel from the most proximal activated neuron, dilatation magnitude decreased and the incidence of constrictions increased. At the highest stimulation level investigated, the incidence of vessel constrictions also increased with cortical depth. With increasing activation levels, capillaries displayed diameter changes that were similar to their immediate neighbors, while vascular network efficiency increased only under the strongest stimulation. Our computational analysis pipeline permits probing microvascular network reactivity and sheds light on the heterogeneity and coordination of vessel caliber changes across the microvascular network. The pipeline will be made available to the research community to propel future studies of neurovascular coupling and network reactivity.

## Acknowledgements

We are grateful to Calcul Québec and the Digital Research Alliance of Canada (alliancecan.ca) for their allocation of compute resources that in part supported this research. The authors wish to thank The Imaging Facility, The Hospital for Sick Children, Toronto, Canada for assistance with light sheet fluorescence microscopy used in collecting data. This work was supported by funding from Canadian Institutes of Health Research grants PJT376309, PJT156179 and PJT178059. MWR was supported by the Queen Elizabeth II/Sunnybrook and Women's College Health Sciences Centre Graduate Scholarships in Science and Technology. BS and MG are supported by the Canada Research Chair program (CRC-2018-00042 and CRC-2021-00374).

## Additional information

### Funding

| Funder | Grant reference number | Author |
| --- | --- | --- |
| Canadian Institutes of Health Research | PJT376309 | Bojana Stefanovic |
| Canadian Institutes of Health Research | PJT156179 | Bojana Stefanovic |

| Funder | Grant reference number | Author |
|---|---|---|
| Canadian Institutes of Health Research | PJT178059 | Bojana Stefanovic |
| Canada Research Chair program | CRC-2021-00374 | Maged Goubran Bojana Stefanovic |
| Canada Research Chair program | CRC-2018-00042 | Maged Goubran Bojana Stefanovic |
| Queen Elizabeth II/ Sunnybrook and Women's College Health Sciences Centre Graduate Scholarships in Science and Technology | | Matthew W Rozak |

The funders had no role in study design, data collection and interpretation, or the decision to submit the work for publication.

## Author contributions

Matthew W Rozak, Conceptualization, Data curation, Software, Formal analysis, Validation, Investigation, Visualization, Methodology, Writing – original draft, Writing – review and editing; James R Mester, Conceptualization, Writing – review and editing; Ahmadreza Attarpour, Software, Writing – review and editing; Adrienne Dorr, Data curation, Methodology, Writing – review and editing; Shruti Patel, Margaret Koletar, Mary E Hill, Data curation; Joanne McLaurin, Resources; Maged Goubran, Conceptualization, Resources, Software, Supervision, Funding acquisition, Validation, Methodology, Writing – original draft, Project administration, Writing – review and editing; Bojana Stefanovic, Conceptualization, Resources, Software, Supervision, Funding acquisition, Validation, Investigation, Visualization, Methodology, Writing – original draft, Project administration, Writing – review and editing

## Author ORCIDs

Matthew W Rozak ![ORCID] https://orcid.org/0009-0001-5604-1405
Ahmadreza Attarpour ![ORCID] https://orcid.org/0000-0001-5890-5740
Adrienne Dorr ![ORCID] https://orcid.org/0009-0004-7628-1678
Shruti Patel ![ORCID] https://orcid.org/0009-0009-5794-4778
Margaret Koletar ![ORCID] https://orcid.org/0000-0003-4464-3887
Mary E Hill ![ORCID] https://orcid.org/0009-0009-1938-4562
Maged Goubran ![ORCID] https://orcid.org/0000-0001-5880-0818
Bojana Stefanovic ![ORCID] https://orcid.org/0000-0002-8439-7601

## Ethics

All experimental procedures in this study followed the ARRIVE 2.0 guidelines (Percie du Sert et al., 2020). They were approved by the Animal Care Committee of the Sunnybrook Research Institute, which adheres to the policies and guidelines of the Canadian Council on Animal Care and meets all the requirements of the Provincial Statute of Ontario, Animals for Research Act, and the Canadian Federal Health of Animals Act.

Reviewer #1 (Public Review): https://doi.org/10.7554/eLife.95525.5.sa1
Reviewer #2 (Public Review): https://doi.org/10.7554/eLife.95525.5.sa2
Author response https://doi.org/10.7554/eLife.95525.5.sa3

---

# Additional files

## Supplementary files

Supplementary file 1. Physiological parameters of the mice during imaging. Includes heart rate, breath rate, and oxygen saturation levels recorded via pulse oximetry.

Supplementary file 2. Deep learning data augmentation parameters. Detailed list of spatial and intensity transformations used for training the segmentation models.

Supplementary file 3. Hyperparameter optimization grid search results. Comparison of model

performance across different loss functions, learning rates, and dropout settings.

Supplementary file 4. Segmentation model performance metrics. Quantitative evaluation of UNETR, U-Net, and ilastik models based on overlap and surface distance metrics.

MDAR checklist

## Data availability

Imaging data used in this paper was deposited in the Federated Research Data Repository (FRDR) and is available at https://doi.org/10.20383/103.01588. The code for the analyses presented in this paper is openly accessible at https://github.com/AICONSlab/novas3d (copy archived at *Rozak and Osmann , 2026*).

The following dataset was generated:

| Author(s) | Year | Dataset title | Dataset URL | Database and Identifier |
|---|---|---|---|---|
| Rozak M, Mester J, Attarpour A, Dorr A, Patel S, Koletar M, Hill M, McLaurin J, Goubran M, Stefanovic B | 2026 | THY1-ChR2-EYFP mouse images used to train and test the NOVAS3D pipeline | https://doi.org/10.20383/103.01588 | Federated Research Data Repository, 10.20383/103.01588 |

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

## Appendix 1

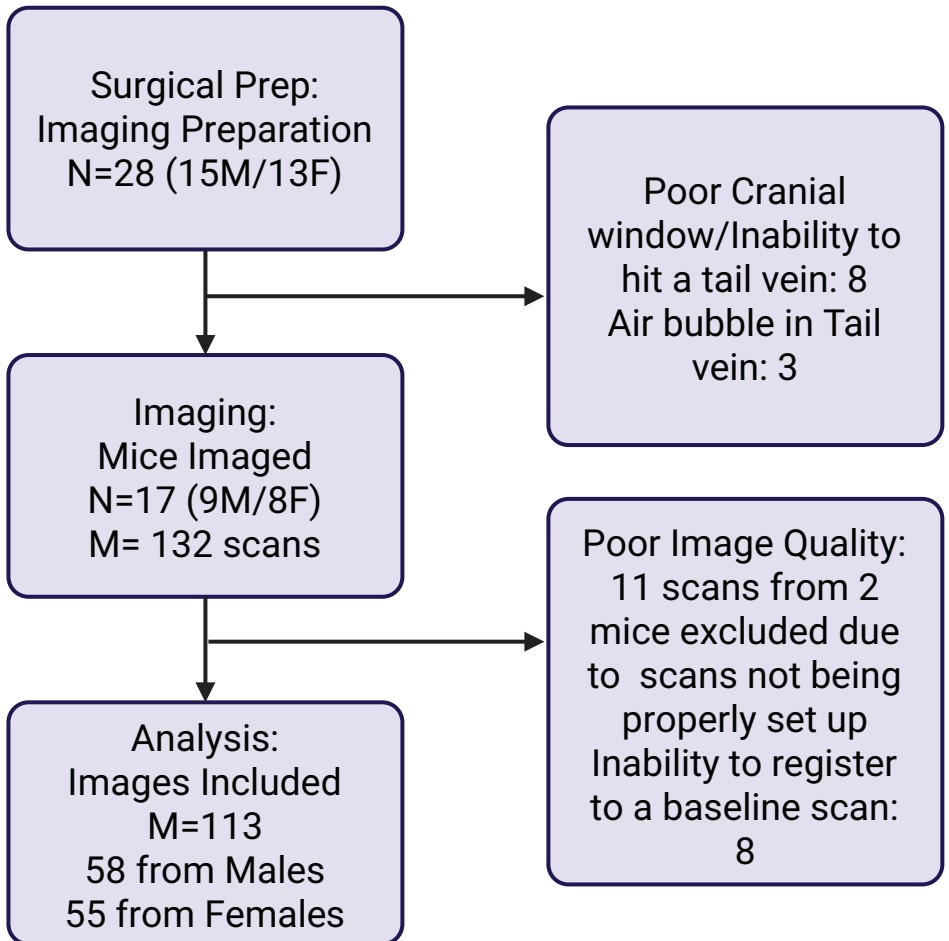

**Appendix 1—figure 1.** Attrition. Flow chart of animal numbers at each step of the experiment.

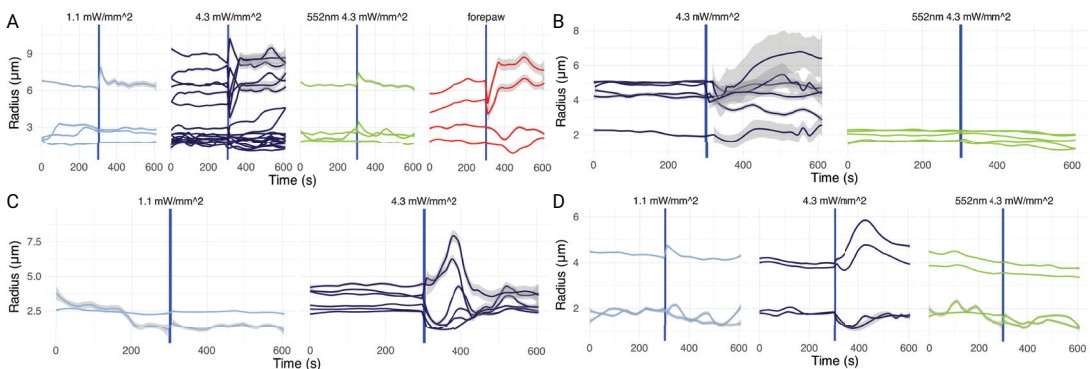

**Appendix 1—figure 2.** High temporal resolution (1.9–3.2 s per frame) time courses of microvascular radii. Loess smoothed vascular radius estimates over time in significantly responding vessels as determined by an F-test comparing the variance in vascular radius before and after stimulation. Light blue corresponds to trials with 1.1 mW/mm$^2$ photostimulation at 458 nm; dark blue, 4.3 mW/mm$^2$ at 458 nm; green, 4.3 mW/mm$^2$ photostimulation at 552 nm; and red, 2 mA, 3 Hz, 10 s on, 10 s off stimulation of the contralateral forepaw. Images were acquired for 5 min prior to stimulation and for 5 min following the end of the stimulation. Optogenetic stimuli were parametrized as described in Imaging. The forepaw stimulation started at the vertical blue line time and lasted for the duration of the scan. (**A**) Sample murine radii traces in vessels whose radius was significantly altered following

*Appendix 1—figure 2 continued on next page*

*Appendix 1—figure 2 continued*

photostimulations. This volume was acquired from a depth of 156–94 µm below the cortical surface with each volume acquisition lasting 2.98 s. (**B**) Another sample murine radii traces in vessels whose radius was significantly altered following photostimulations. This volume was acquired from a depth of 75–5 µm below the cortical surface with each volume acquisition lasting 3.17 s. (**C, D**) Traces of responding vessel radii from the third mouse. (**C**) This volume was acquired from a depth of 305–341 µm below the cortical surface with each volume acquisition lasting 1.93 s. (**D**) This volume was acquired from a depth of 276–235 µm below the cortical surface with each volume acquisition lasting 2.16 s.

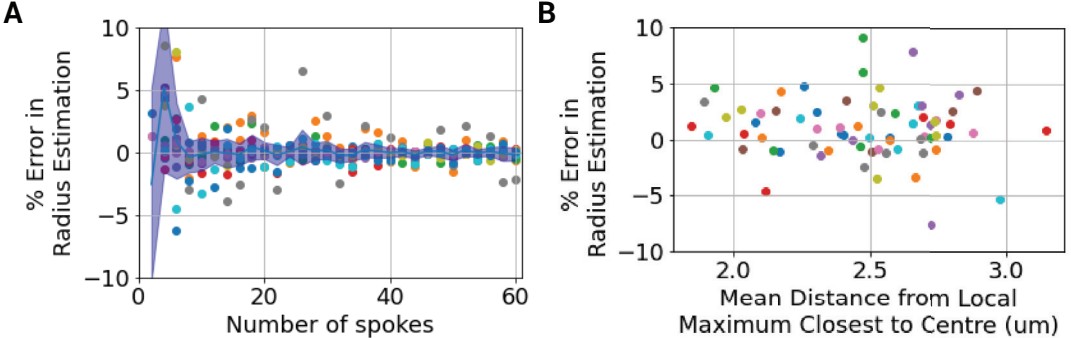

**Appendix 1—figure 3.** Bead radius estimation sensitivity to noise. (**A**) Sensitivity of vessel-wise radius estimate on the number of spokes used to estimate the radius. The radius estimate converges after 20 spokes have been used for estimation. Using 36 spokes initially, the vesselwise mean radius estimation was within 0.24±0.62% of the mean of radius estimates using 40–60 spokes. (**B**) The centerline was jittered in the perpendicular plane at each point along the line and then mean radius was estimated in 71 larger vessels (mean radius > 5 µm). The percent difference in the estimated radius at our selected vessel centerpoints vs. the jittered centerpoints is plotted. The percent difference in the mean radius estimation was 0.64 ± 3.44% (eg. 0.032±0.17 µm for a vessel with a radius of 5 µm) with 2.45±0.30 µm centerline jittering. (In contrast, photostimulation was estimated to elicit an average 25.4 ± 18.1% change in the magnitude of the radius of larger vessels, that is those with the baseline radius > 5 µm.).

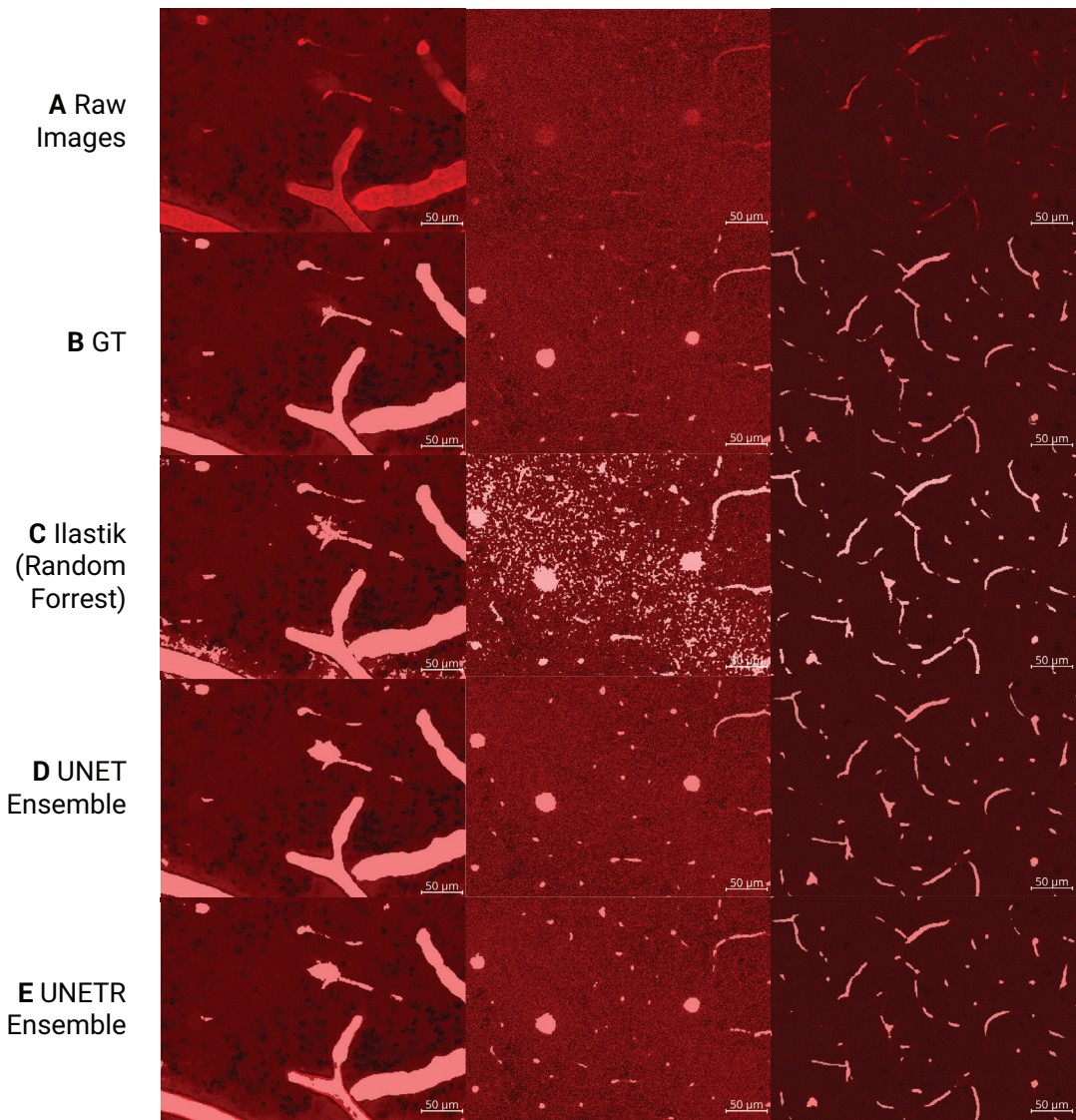

**Appendix 1—figure 4.** Sample 2D slices of segmentation results. (**A**) Raw slices of the vascular channel with the neuron channel subtracted to facilitate vessel visualization. The first slice was 29 µm below the cortical surface; the second slice 200 µm; and the third image 300 µm. Each slice is from a separate mouse. All images were taken from the test dataset, unseen during model training. (**B**) Ground truth segmentation masks for the vasculature were generated by a rater who utilized ilastik assisted manual segmentation. (**C**) Ilastik predictions generated via a random forest model. (**D**) Binary segmentation masks generated by an ensemble of 3D UNet models. (**E**) Binary segmentation masks generated by an ensemble of 3D UNETR models.

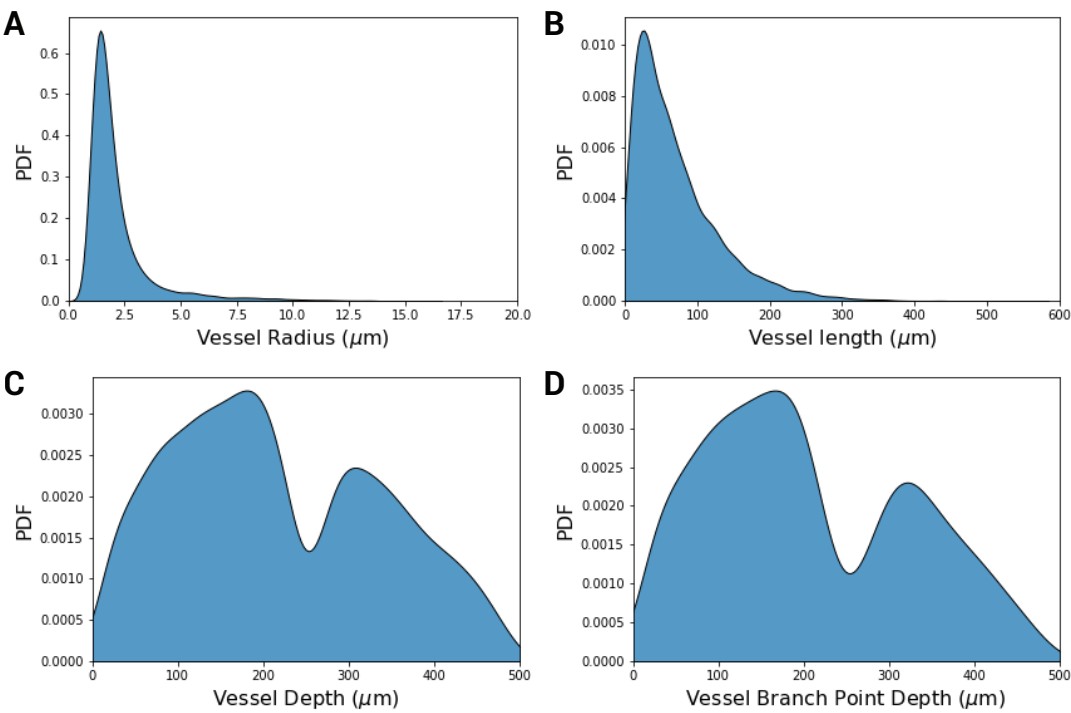

**Appendix 1—figure 5.** Vascular network characteristics. (**A**) Probability density of the extracted mean radii of vascular segments. (**B**) Probability density of the lengths of extracted vessel segments between branch points or terminal ends. (**C**) Probability density of the mean vessel segment depths. (**D**) Probability density of the depths of vessel branch points. Terminal nodes were excluded from this probability density. 12,555 vessels and 6421 vascular junctions from 17 mice (9M/8F) were used to estimate these PDFs.

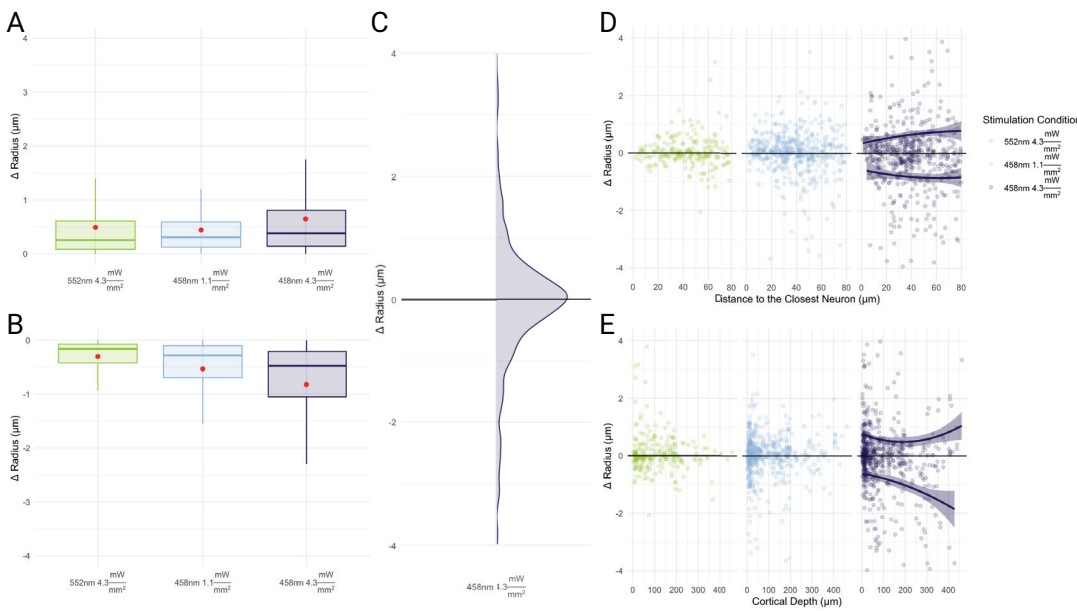

**Appendix 1—figure 6.** Large vessels' radius changes following optogenetic stimulation. All vascular responses in large vessels (radius > 5 µm) separated into dilators (**A**) and constrictors (**B**) showing an increase in the magnitude of the vascular response as the photostimulation power increases. (**C**) Probability density function of diameter changes across all large vessels. (**D**) Radius changes to vascular radii in relation to the closest pyramidal neurons (within 80 µm). (**E**) Mean depth of the responding large vessels.

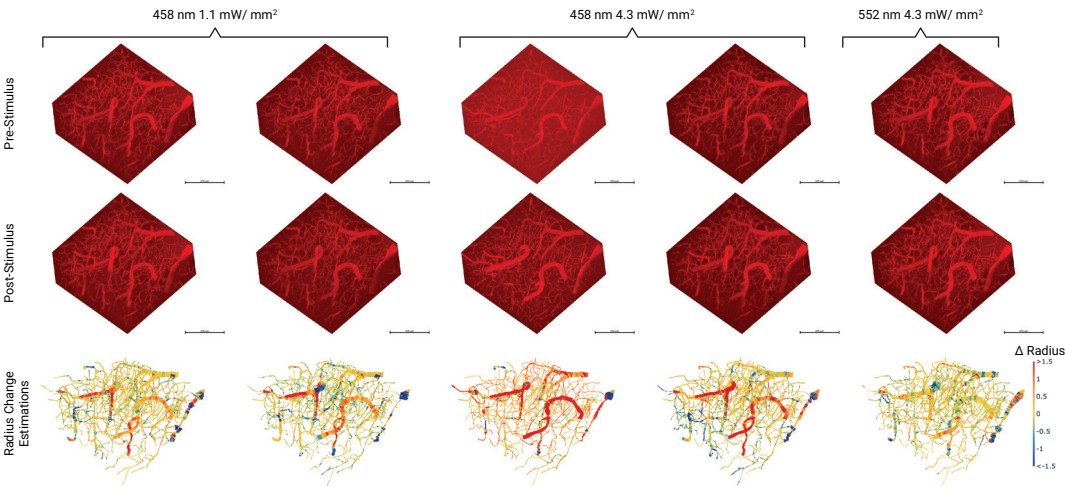

**Appendix 1—figure 7.** Registered images of the cortical microvasculature before and after optogenetic stimulation for five scan pairs over three different stimulation conditions. The estimated radii changes along vessel segments are shown in the third row.

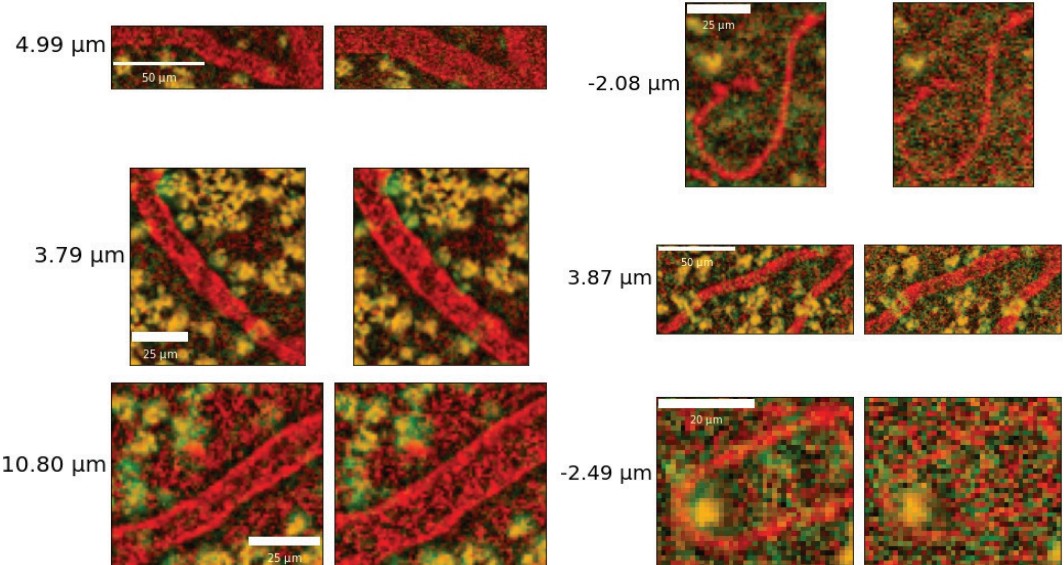

**Appendix 1—figure 8.** Maximum intensity projections of sample capillary constrictions at repeated time points following optogenetic stimulation. Baseline (pre-stimulation) image is shown on the left and the post-stimulation image, on the right, with the estimated radius changes listed to the left.

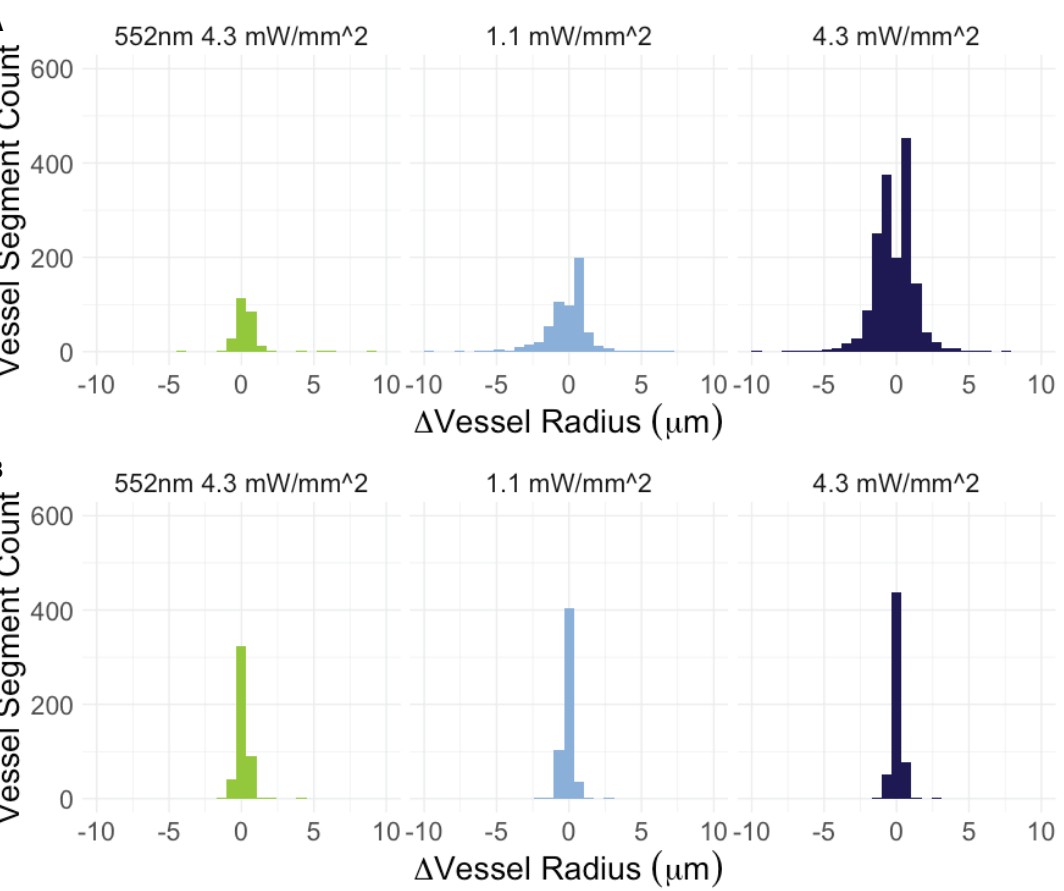

**Appendix 1—figure 9.** THY1-ChR2 photostimulation-induced changes in vessel-wise microvascular radii compared to C57BL/6J. (**A**) Vessel radius in responding vessels of the Thy1-ChR2 mice described in the manuscript vs. (**B**) Four wild-type C57BL/6J mice. Response to photostimulation was defined as a radius change above twice the standard deviation in the radius across baseline frames. 552 nm light was applied at 4.3 mW/mm$^2$, while 458 nm light was applied at 1.1 mW/mm$^2$ and 4.3 mW/mm$^2$. In C57BL/6J mice (**B**), the radii distributions following either blue light photostimulation were not statistically distinguishable from that resulting from green photostimulation ('response' to control condition) using a Wilcoxon test.

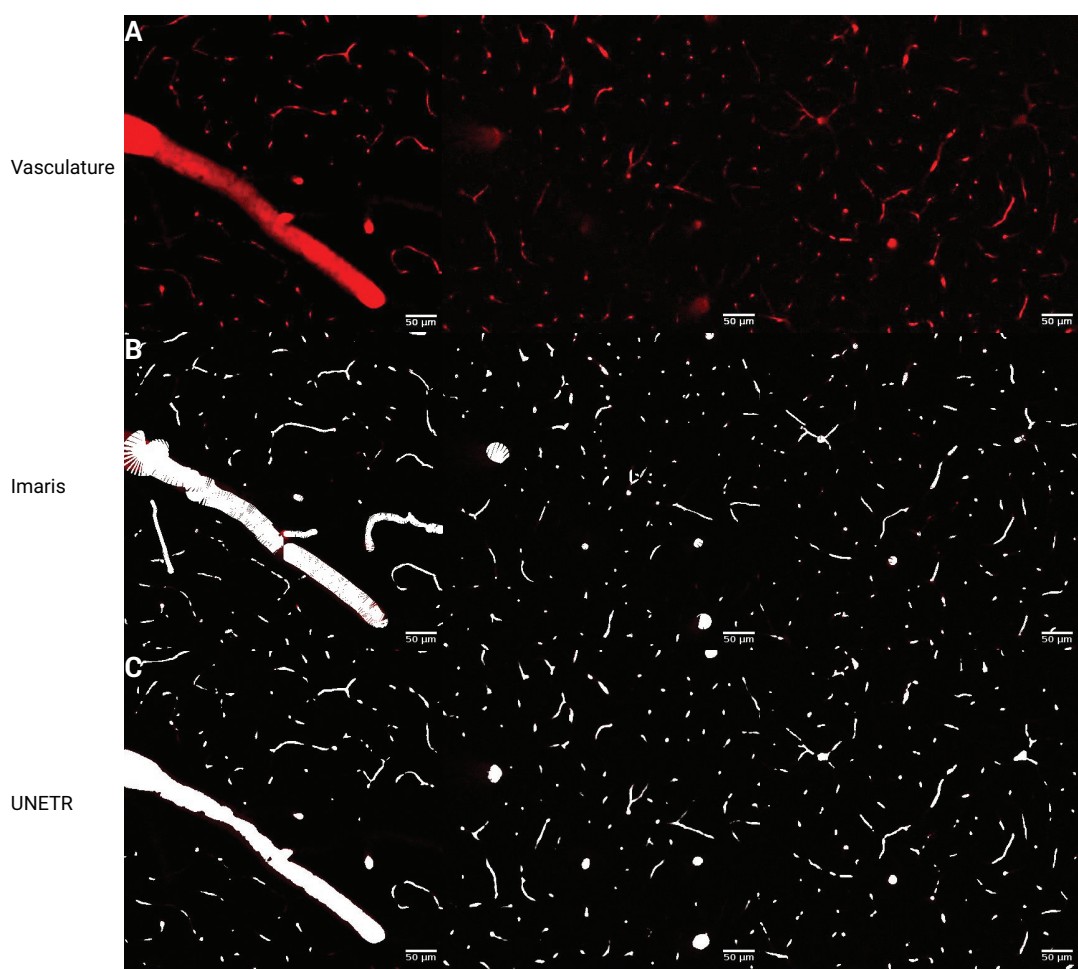

**Appendix 1—figure 10.** Imaris segmentation examples. (**A**) Raw slices of the vascular channel with the neuronal channel subtracted to facilitate vessel visualization. (**B**) Corresponding slices of the segmentation mask generated in Imaris 9.2.1 manually by a rater overlaid onto the raw data. (**C**) Corresponding segmentation masks generated by our UNETR segmentation model overlaid onto the raw data.

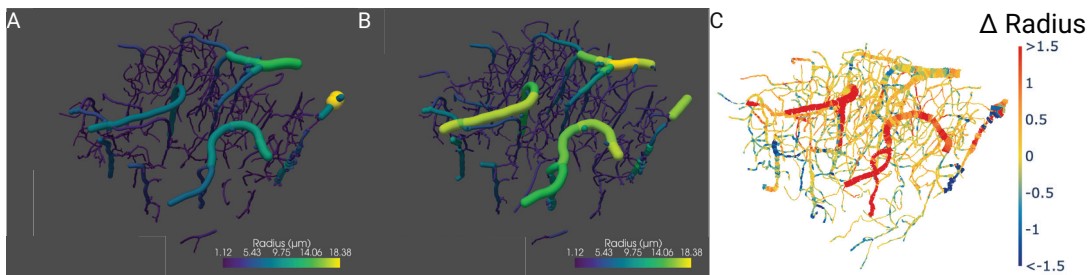

**Appendix 1—figure 11.** Results of VesselVio graph generation on our segmentation masks. Graphs were generated on the images shown in column 4 of *Appendix 1—figure 7*. (**A**) VesselVio vascular graph extraction on a volume before blue light photostimulation produced 546 vessel segments. (**B**) VesselVio vascular graph extraction on the same imaging volume after blue light stimulation produced 642 vessel segments. (**C**) NOVAS3D generated graph of the vasculature with direct tracking of morphological changes.

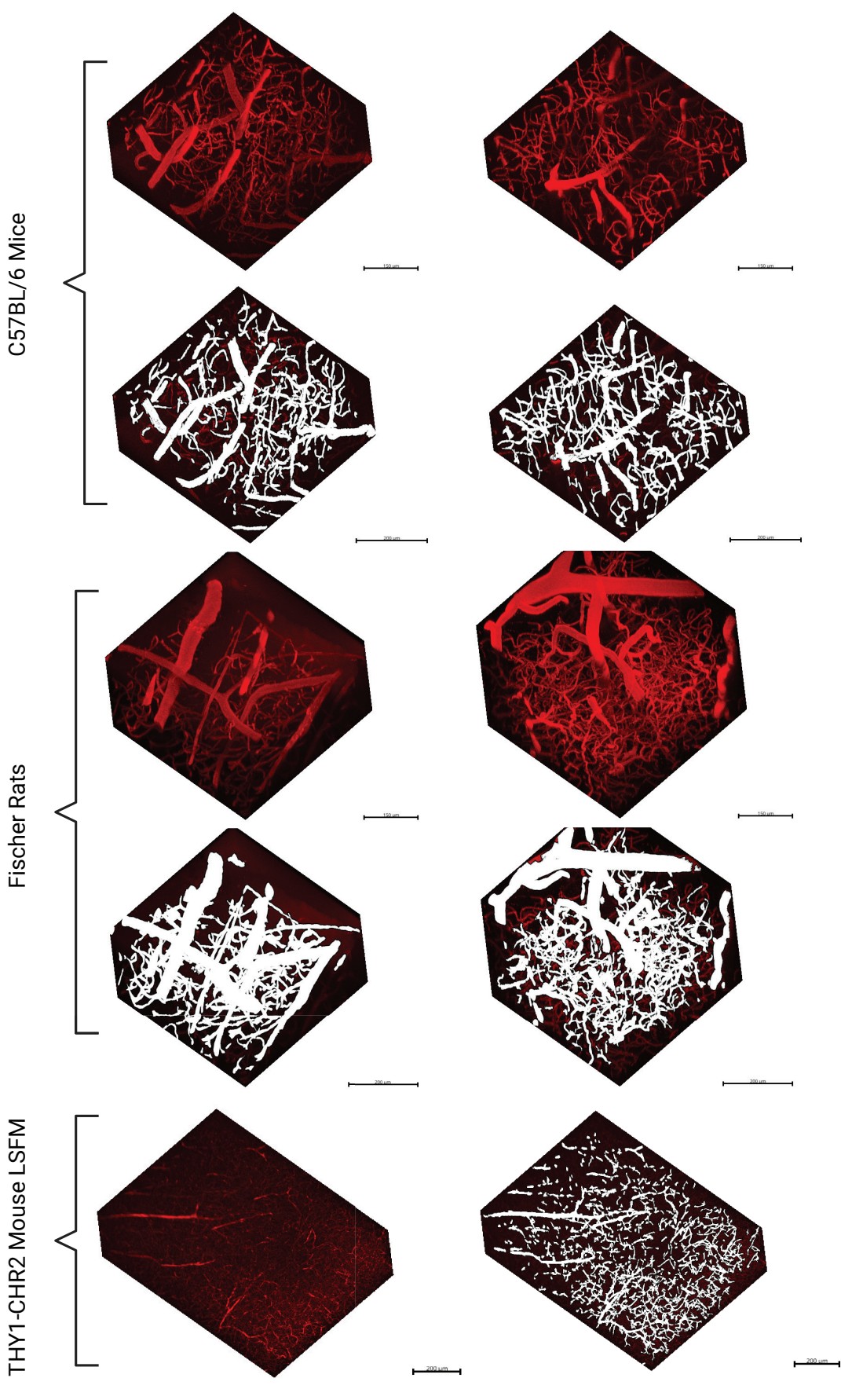

**Appendix 1—figure 12.** Examples of the predictions of the deep learning model applied on out-of-distribution data from a different mouse strain (C57BL/6J), a different species (Fischer rat), and a different microscope (light sheet fluorescence microscope, Miltenyi UltraMicroscope Blaze).

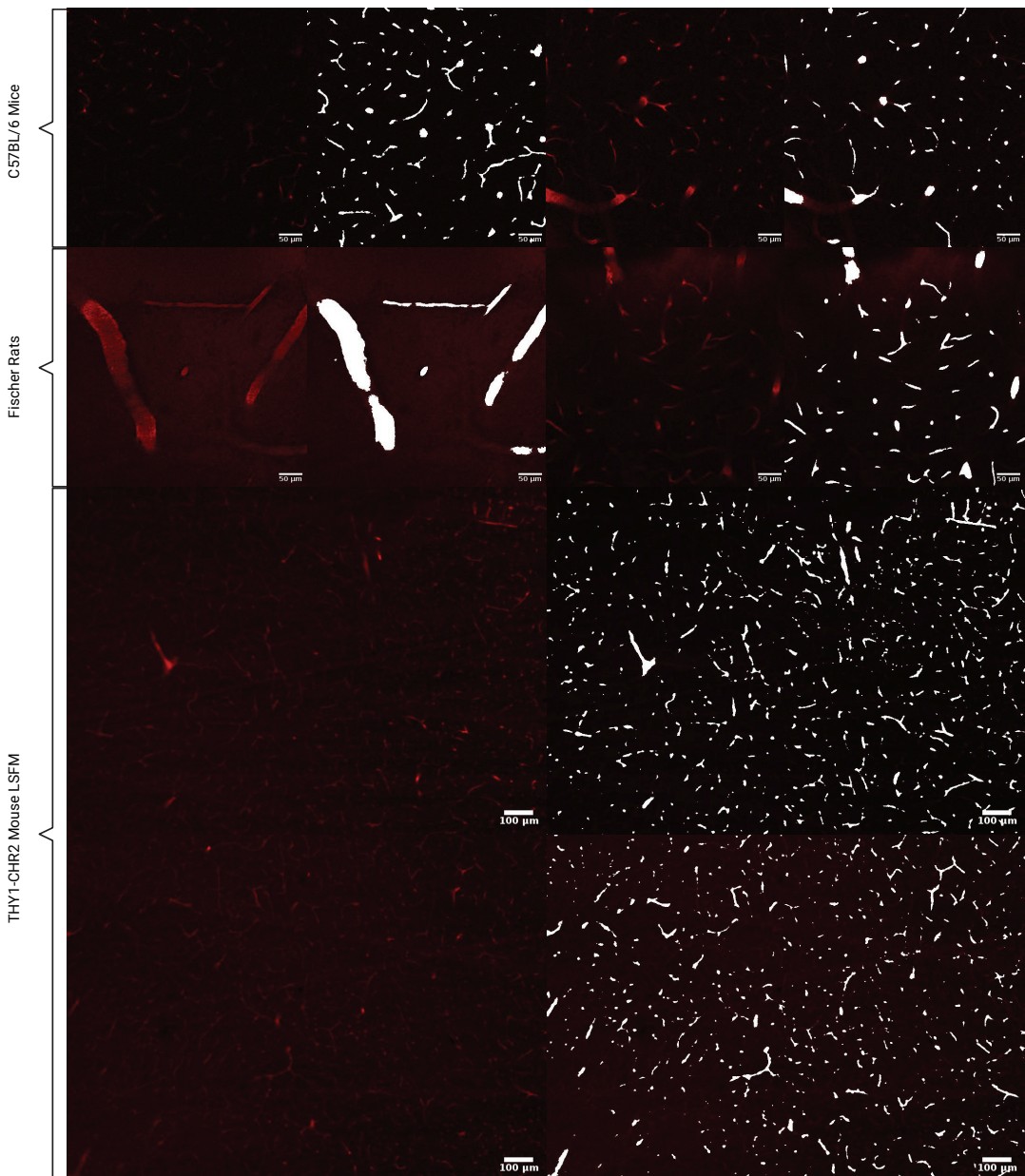

**Appendix 1—figure 13.** Examples of 2D slices of the predictions of the NOVAS3D deep learning model applied on out-of-distribution data from a different mouse strain (C57BL/6), a different species (Fischer rat), and a different microscope (light sheet fluorescence microscope, Miltenyi UltraMicroscope Blaze).

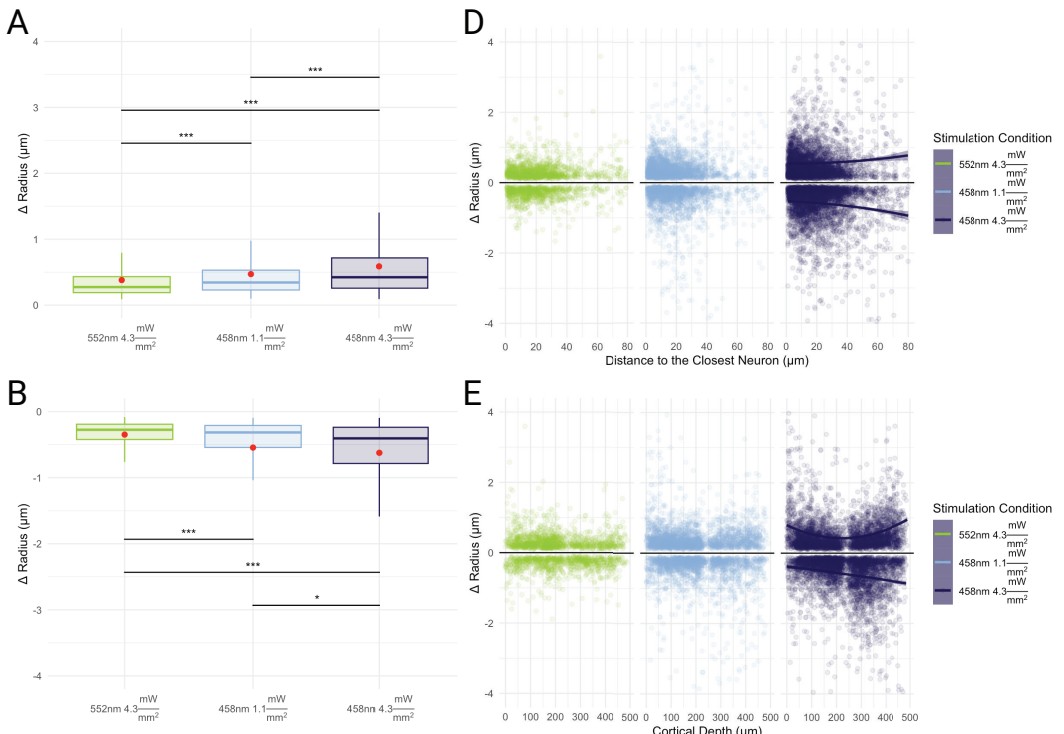

**Appendix 1—figure 14.** Optogenetic activation-induced changes in vessel-wise microvascular radii with responders defined as vessels changing their radius by more than 10%. Capillary responses included both dilatations, shown in (**A**), and constrictions, shown in (**B**) with potentiation of the capillary response with increased photostimulation power. * p<0.05, ** p<0.005, and *** p<0.0005. p-values were not adjusted. (**C**) Changes to capillary radii are displayed in relation to the closest labeled neuron. The proportion of vessels constricting increased with the higher intensity of blue light stimulation, and constrictions tended to occur further away from labeled neurons than did dilations. (**D**) Mean cortical depth of responding capillaries showed a tendency for dilators to be closer to the surface and for constrictors to be deeper in the tissue.

