## [Editor Report · eLife Assessment]

This work describes a highly complex automated algorithm for analyzing vascular imaging data from two-photon microscopy. This tool has the potential to be extremely **valuable** to the field and to fill gaps in knowledge of hemodynamic activity across a regional network. The **solid** biological application provides a demonstration of their pipeline's capabilities and suggests intriguing hypotheses around prolonged vascular tone changes, but will need to be followed up by further experiments to be conclusively demonstrated.

---

## [Referee Report · Reviewer #1 (Public Review)]

Summary:

In this manuscript, the authors describe a new pipeline to measure changes in vasculature diameter upon opt-genetic stimulation of neurons.

The work is interesting and the topic is quite relevant to better understand the hemodynamic response on the graph/network level.

Strengths:

The manuscript provides a pipeline that allows for the detection of changes in the vessel diameter as well as simultaneously allowing for the location of the neurons driven by stimulation.

The resulting data could provide interesting insights into the graph-level mechanisms of regulating activity-dependent blood flow.

The interesting findings include that vessel radius changes depend on depth from the cortical surface and that dilations on average happen closer to the activated neurons.

---

## [Referee Report · Reviewer #2 (Public Review)]

Summary:

The authors develop a highly detailed pipeline to analyze hemodynamic signals from in vivo two-photon fluorescence microscopy. This includes motion correction, segmentation of the vascular network, diameter measurements across time, mapping neuronal position relative to the vascular network, and analyzing vascular network properties (interactions between different vascular segments). For the segmentation, the authors use a Convolution Neural Network to identify vessel (or neural) and background pixels and train it using ground truth images based on semi-automated mapping followed by human correction/annotation. Considerable processing was done on the segmented images to improve accuracy, extract vessel center lines, and compute frame-by-frame diameters. The model was tested with artificial diameter increases and Gaussian noise and proved robust to these manipulations.

Network-level properties include Assortativity - a measure of how similar a vessel's response is to nearby vessels - and Efficiency - the ease of flow through the network (essentially, the combined resistance of a path based on diameter and vessel length between two points).

Strengths:

This is a very powerful tool for cerebral vascular biologists as many of these tasks are labor intensive, prone to subjectivity, and often not performed due to the complexity of collecting and managing volumes of vascular signals. Modelling is not my specialty so I cannot speak too specifically, but the model appears to be well-designed and robust to perturbations. It has many clever features for processing the data.

The authors rightly point out that there is a real lack in the field of knowledge of vascular network activity at single-vessel resolution. Network anatomy has been studied, but hemodynamics are typically studied either with coarse resolution or in only one or a few vessels at a time. This pipeline has the potential to change that.

[Editors' note: this work has been through three rounds of revisions, and most recently the authors have added caveats to the discussion. This version of the paper has been assessed by the editors and the weaknesses identified previously remain with earlier versions of the work.]

---

## [Author Response]

The following is the authors’ response to the previous reviews

**Reviewer #1 (Public review):**
Summary:In the manuscript the authors describe a new pipeline to measure changes in vasculature diameter upon optogenetic stimulation of neurons. The work is useful to better understand the hemodynamic response on a network /graph level.Strengths:The manuscript provides a pipeline that allows to detect changes in the vessel diameter as well as simultaneously allows to locate the neurons driven by stimulation.The resulting data could provide interesting insights into the graph level mechanisms of regulating activity dependent blood flow.Weaknesses:(1) The manuscript contains (new) wrong statements and (still) wrong mathematical formulas.

The symbols in these formulas have been updated to disambiguate them, and the accompanying statements have been adjusted for clarity.

(2) The manuscript does not compare results to existing pipelines for vasculature segmentation (opensource or commercial). Comparing performance of the pipeline to a random forest classifier (illastik) on images that are not preprocessed (i.e. corrected for background etc.) seems not a particularly useful comparison.

We’ve now included comparisons to Imaris (a commercial) for segmentation and VesselVio (open-source) for graph extraction software.

For the ilastik comparison, the images were preprocessed prior to ilastik segmentation, specifically by doing intensity normalization.

Example segmentations utilizing Imaris have now been included. Imaris leaves gaps and discontinuities in the segmentation masks, as shown in Supplementary Figure 10. The Imaris segmentation masks also tend to be more circular in cross-section despite irregularities on the surface of the vessels observable in the raw data and identified in manual segmentation. This approach also requires days to months to generate per image stack.

A comparison to VesselVio has now also been generated, and results are visualized in Supplementary Figure 11. VesselVio generates individual graphs for each time point, resulting in potential discrepancies in the structure of the graphs from different time points. Furthermore, Vesselvio uses distance transformation to estimate the vascular radius, which renders the vessel radius estimates highly susceptible to variation in the user selected methodology used to obtain segmentation results; while our approach uses intensity gradient-based boundary detection from centerlines in the image instead mitigating this bias. We have added the following paragraph to the Discussion section on the comparisons with the two methods:

“Comparison with commercial and open-source vascular analysis pipelines

To compare our results with those achievable on these data with other pipelines for segmentation and graph network extraction, we compared segmentation results qualitatively with Imaris version 9.2.1 (Bitplane) and vascular graph extraction with VesselVio [1]**.** For the Imaris comparison, three small volumes were annotated by hand to label vessels. Example slices of the segmentation results are shown in Supplementary Figure 10. Imaris tended to either over- or under-segment vessels, disregard fine details of the vascular boundaries, and produce jagged edges in the vascular segmentation masks. In addition to these issues with segmentation mask quality, manual segmentation of a single volume took days for a rater to annotate. To compare to VesselVio, binary segmentation masks (one before and one after photostimulation) generated with our deep learning models were loaded into VesselVio for graph extraction, as VesselVio does not have its own method for generating segmentation masks. This also facilitates a direct comparison of the benefits of our graph extraction pipeline to VesselVio. Visualizations of the two graphs are shown in Supplementary Figure 11. Vesselvio produced many hairs at both time points, and the total number of segments varied considerably between the two sequential stacks: while the baseline scan resulted in 546 vessel segments, the second scan had 642 vessel segments. These discrepancies are difficult to resolve in post-processing and preclude a direct comparison of individual vessel segments across time. As the segmentation masks we used in graph extraction derive from the union of multiple time points, we could better trace the vasculature and identify more connections in our extracted graph. Furthermore, VesselVio relies on the distance transform of the user supplied segmentation mask to estimate vascular radii; consequently, these estimates are highly susceptible to variations in the input segmentation masks.We repeatedly saw slight variations between boundary placements of all of the models we utilized (ilastik, UNet, and UNETR) and those produced by raters. Our pipeline mitigates this segmentation method bias by using intensity gradient-based boundary detection from centerlines in the image (as opposed to using the distance transform of the segmentation mask, as in VesselVio).”

(3) The manuscript does not clearly visualize performance of the segmentation pipeline (e.g. via 2d sections, highlighting also errors etc.). Thus, it is unclear how good the pipeline is, under what conditions it fails or what kind of errors to expect.

On reviewer’s comment, 2D slices have been added in the Supplementary Figure 4.

(4) The pipeline is not fully open-source due to use of matlab. Also, the pipeline code was not made available during review contrary to the authors claims (the provided link did not lead to a repository). Thus, the utility of the pipeline was difficult to judge.

All code has been uploaded to Github and is available at the following location: https://github.com/AICONSlab/novas3d

The Matlab code for skeletonization is better at preserving centerline integrity during the pruning of hairs from centerlines than the currently available open-source methods.

- Generalizability: The authors addressed the point of generalizability by applying the pipeline to other data sets. This demonstrates that their pipeline can be applied to other data sets and makes it more useful. However, from the visualizations it's unclear to see the performance of the pipeline, where the pipelines fails etc. The 3d visualizations are not particularly helpful in this respect . In addition, the dice measure seems quite low, indicating roughly 20-40% of voxels do not overlap between inferred and ground truth. I did not notice this high discrepancy earlier. A thorough discussion of the errors appearing in the segmentation pipeline would be necessary in my view to better assess the quality of the pipeline.

2D slices from the additional datasets have been added in the Supplementary Figure 13 to aid in visualizing the models’ ability to generalize to other datasets.

The dice range we report on (0.7-0.8) is good when compared to those (0.56-86) of 3D segmentations of large datasets in microscopy [2], [3], [4], [5], [6]. Furthermore, we had two additional raters segment three images from the original training set. We found that the raters had a mean inter class correlation of 0.73 [7]. Our model outperformed this Dice score on unseen data: Dice scores from our generalizability tests on C57 mice and Fischer rats on par or higher than this baseline.

**Reviewer #2 (Public review):**
The authors have addressed most of my concerns sufficiently. There are still a few serious concerns I have. Primarily, the temporal resolution of the technique still makes me dubious about nearly all of the biological results. It is good that the authors have added some vessel diameter time courses generated by their model. But I still maintain that data sampling every 42 seconds - or even 21 seconds - is problematic. First, the evidence for long vascular responses is lacking. The authors cite several papers:Alarcon-Martinez et al. 2020 show and explicitly state that their responses (stimulus-evoked) returned to baseline within 30 seconds. The responses to ischemia are long lasting but this is irrelevant to the current study using activated local neurons to drive vessel signals.Mester et al. 2019 show responses that all seem to return to baseline by around 50 seconds post-stimulus.

In Mester et al. 2019, diffuse stimulations with blue light showed a return to baseline around 50 seconds post-stimulus (Figure 1E,2C,2D). However, focal stimulations where the stimulation light is raster scanned over a small region focused in the field of view show longer-lasting responses (Figure 4) that have not returned to baseline by 70 seconds post-stimulus [8]. Alarcon-Martinez et al. do report that their responses return baseline within 30 seconds; however, their physiological stimulation may lead to different neuronal and vessel response kinetics than those elicited by the optogenetic stimulations as in current work.

O'Herron et al. 2022 and Hartmann et al. 2021 use opsins expressed in vessel walls (not neurons as in the current study) and directly constrict vessels with light. So this is unrelated to neuronal activity-induced vascular signals in the current study.

We agree that optogenetic activation of vessel-associated cells is distinct from optogenetic activation of neurons, but we do expect the effects of such perturbations on the vasculature to have some commonalities.

There are other papers including Vazquez et al 2014 (PMID: 23761666) and Uhlirova et al 2016 (PMID: 27244241) and many others showing optogenetically-evoked neural activity drives vascular responses that return back to baseline within 30 seconds. The stimulation time and the cell types labeled may be different across these studies which can make a difference. But vascular responses lasting 300 seconds or more after a stimulus of a few seconds are just not common in the literature and so are very suspect - likely at least in part due to the limitations of the algorithm.

The photostimulation in Vazquez et al. 2014 used diffuse photostimulation with a fiberoptic probe similar to Mester et al. 2019 as opposed to raster scanning focal stimulation we used in this study and in the study by Mester et al. 2019 where we observed the focal photostimulation to elicited longer than a minute vascular responses. Uhlirova et al. 2016 used photostimulation powers between 0.7 and 2.8 mW, likely lower than our 4.3 mW/mm^2^ photostimulation. Further, even with focal photostimulation, we do see light intensity dependence of the duration of the vascular responses. Indeed, in Supplementary Figure 2, 1.1 mW/mm^2^ photostimulation leads to briefer dilations/constrictions than does 4.3 mW/mm^2^; the 1.1 mW/mm^2^ responses are in line, duration wise, with those in Uhlirova et al. 2016.

Critically, as per Supplementary Figure 2, the analysis of the experimental recordings acquired at 3-second temporal resolution did likewise show responses in many vessels lasting for tens of seconds and even hundreds of seconds in some vessels.

Another major issue is that the time courses provided show that the same vessel constricts at certain points and dilates later. So where in the time course the data is sampled will have a major effect on the direction and amplitude of the vascular response. In fact, I could not find how the "response" window is calculated. Is it from the first volume collected after the stimulation - or an average of some number of volumes? But clearly down-sampling the provided data to 42 or even 21 second sampling will lead to problems. If the major benefit to the field is the full volume over large regions that the model can capture and describe, there needs to be a better way to capture the vessel diameter in a meaningful way.

In the main experiment (i.e. excluding the additional experiments presented in the Supplementary Figure 2 that were collected over a limited FOV at 3s per stack), we have collected one stack every 42 seconds. The first slice of the volume starts following the photostimulation, and the last slice finishes at 42 seconds. Each slice takes ~0.44 seconds to acquire. The data analysis pipeline (as demonstrated by the Supplementary Figure 2) is not in any way limited to data acquired at this temporal resolution and - provided reasonable signal-to-noise ratio (Figure 5) - is applicable, as is, to data acquired at much higher sampling rates.

It still seems possible that if responses are bi-phasic, then depth dependencies of constrictors vs dilators may just be due to where in the response the data are being captured - maybe the constriction phase is captured in deeper planes of the volume and the dilation phase more superficially. This may also explain why nearly a third of vessels are not consistent across trials - if the direction the volume was acquired is different across trials, different phases of the response might be captured.

Alternatively, like neuronal responses to physiological stimuli, the vascular responses elicited by increases in neuronal activity may themselves be variable in both space and time.

I still have concerns about other aspects of the responses but these are less strong. Particularly, these bi-phasic responses are not something typically seen and I still maintain that constrictions are not common. The authors are right that some papers do show constriction. Leaving out the direct optogenetic constriction of vessels (O'Herron 2022 & Hartmann 2021), the Alarcon-Martinez et al. 2020 paper and others such as Gonzales et al 2020 (PMID: 33051294) show different capillary branches dilating and constricting. However, these are typically found either with spontaneous fluctuations or due to highly localized application of vasoactive compounds. I am not familiar with data showing activation of a large region of tissue - as in the current study - coupled with vessel constrictions in the same region. But as the authors point out, typically only a few vessels at a time are monitored so it is possible - even if this reviewer thinks it unlikely - that this effect is real and just hasn't been seen.

Uhlirova et al. 2016 (PMID: 27244241) observed biphasic responses in the same vessel with optogenetic stimulation in anesthetized and unanesthetized animals (Fig 1b and Fig 2, and section “OG stimulation of INs reproduces the biphasic arteriolar response”). Devor et al. (2007) and Lindvere et al. (2013) also reported on constrictions and dilations being elicited by sensory stimuli.

I also have concerns about the spatial resolution of the data. It looks like the data in Figure 7 and Supplementary Figure 7 have a resolution of about 1 micron/pixel. It isn't stated so I may be wrong. But detecting changes of less than 1 micron, especially given the noise of an in vivo prep (brain movement and so on), might just be noise in the model. This could also explain constrictions as just spurious outputs in the model's diameter estimation. The high variability in adjacent vessel segments seen in Figure 6C could also be explained the same way, since these also seem biologically and even physically unlikely.

Thank you for your comment. To address this important issue, we performed an additional validation experiment where we placed a special order of fluorescent beads with a known diameter of 7.32 ± 0.27um, imaged them following our imaging protocol, and subsequently used our pipeline to estimate their diameter. Our analysis converged on the manufacturer-specified diameters, estimating the diameter to be 7.34 ± 0.32. The manuscript has been updated to detail this experiment, as below:

Methods section insert

“Second, our boundary detection algorithm was used to estimate the diameters of fluorescent beads of a known radius imaged under similar acquisition parameters. Polystyrene microspheres labelled with Flash Red (Bangs Laboratories, inc, CAT# FSFR007) with a nominal diameter of 7.32um and a specified range of 7.32 ± 0.27um as determined by the manufacturer using a Coulter counter were imaged on the same multiphoton fluorescence microscope set-up used in the experiment (identical light path, resonant scanner, objective, detector, excitation wavelength and nominal lateral and axial resolutions, with 5x averaging). The images of the beads had a higher SNR than our images of the vasculature, so Gaussian noise was added to the images to degrade the SNR to the same level of that of the blood vessels. The images of the beads were segmented with a threshold, centroids calculated for individual spheres, and planes with a random normal vector extracted from each bead and used to estimate the diameter of the beads. The same smoothing and PSF deconvolution steps were applied in this task. We then reported the mean and standard deviation of the distribution of the diameter estimates. A variety of planes were used to estimate the diameters.”

Results Section Insert

“Our boundary detection algorithm successfully estimated the radius of precisely specified fluorescent beads. The bead images had a signal-to-noise ratio of 6.79 ± 0.16 (about 35% higher than our in vivo images): to match their SNR to that of in vivo vessel data, following deconvolution, we added Gaussian noise with a standard deviation of 85 SU to the images, bringing the SNR down to 5.05 ± 0.15. The data processing pipeline was kept unaltered except for the bead segmentation, performed via image thresholding instead of our deep learning model (trained on vessel data). The bead boundary was computed following the same algorithm used on vessel data: i.e., by the average of the minimum intensity gradients computed along 36 radial spokes emanating from the centreline vertex in the orthogonal plane. To demonstrate an averaging-induced decrease in the uncertainty of the bead radius estimates on a scale that is finer than the nominal resolution of the imaging configuration, we tested four averaging levels in 289 beads. Three of these averaging levels were lower than that used on the vessels, and one matched that used on the vessels (36 spokes per orthogonal plane and a minimum of 10 orthogonal planes per vessel). As the amount of averaging increased, the uncertainty on the diameter of the beads decreased, and our estimate of the bead's diameter converged upon the manufacturer's Coulter counter-based specifications (7.32 ± 0.27um), as tabulated below in Table 1.”

Bibliography

(1) J. R. Bumgarner and R. J. Nelson, “Open-source analysis and visualization of segmented vasculature datasets with VesselVio,” Cell Rep. Methods, vol. 2, no. 4, Apr. 2022, doi: 10.1016/j.crmeth.2022.100189.

(2) G. Tetteh et al., “DeepVesselNet: Vessel Segmentation, Centerline Prediction, and Bifurcation Detection in 3-D Angiographic Volumes,” Front. Neurosci., vol. 14, Dec. 2020, doi: 10.3389/fnins.2020.592352.

(3) N. Holroyd, Z. Li, C. Walsh, E. Brown, R. Shipley, and S. Walker-Samuel, “tUbe net: a generalisable deep learning tool for 3D vessel segmentation,” Jul. 24, 2023, bioRxiv. doi: 10.1101/2023.07.24.550334.

(4) W. Tahir et al., “Anatomical Modeling of Brain Vasculature in Two-Photon Microscopy by Generalizable Deep Learning,” BME Front., vol. 2020, p. 8620932, Dec. 2020, doi: 10.34133/2020/8620932.

(5) R. Damseh, P. Delafontaine-Martel, P. Pouliot, F. Cheriet, and F. Lesage, “Laplacian Flow Dynamics on Geometric Graphs for Anatomical Modeling of Cerebrovascular Networks,” ArXiv191210003 Cs Eess Q-Bio, Dec. 2019, Accessed: Dec. 09, 2020. (Online). Available: http://arxiv.org/abs/1912.10003

(6) T. Jerman, F. Pernuš, B. Likar, and Ž. Špiclin, “Enhancement of Vascular Structures in 3D and 2D Angiographic Images,” IEEE Trans. Med. Imaging, vol. 35, no. 9, pp. 2107–2118, Sep. 2016, doi: 10.1109/TMI.2016.2550102.

(7) T. B. Smith and N. Smith, “Agreement and reliability statistics for shapes,” PLOS ONE, vol. 13, no. 8, p. e0202087, Aug. 2018, doi: 10.1371/journal.pone.0202087.

(8) J. R. Mester et al., “In vivo neurovascular response to focused photoactivation of Channelrhodopsin-2,” NeuroImage, vol. 192, pp. 135–144, May 2019, doi: 10.1016/j.neuroimage.2019.01.036.